# Wide Neural Networks of Any Depth Evolve as Linear Models Under Gradient Descent

**Jaehoon Lee**[*], **Lechao Xiao**[*], **Samuel S. Schoenholz**, **Yasaman Bahri**
**Roman Novak**, **Jascha Sohl-Dickstein**, **Jeffrey Pennington**

Google Brain

{jaehlee, xlc, schsam, yasamanb, romann, jaschasd, jpennin}@google.com

## Abstract

A longstanding goal in deep learning research has been to precisely characterize training and generalization. However, the often complex loss landscapes of neural networks have made a theory of learning dynamics elusive. In this work, we show that for wide neural networks the learning dynamics simplify considerably and that, in the infinite width limit, they are governed by a linear model obtained from the first-order Taylor expansion of the network around its initial parameters. Furthermore, mirroring the correspondence between wide Bayesian neural networks and Gaussian processes, gradient-based training of wide neural networks with a squared loss produces test set predictions drawn from a Gaussian process with a particular compositional kernel. While these theoretical results are only exact in the infinite width limit, we nevertheless find excellent empirical agreement between the predictions of the original network and those of the linearized version even for finite practically-sized networks. This agreement is robust across different architectures, optimization methods, and loss functions.

## 1   Introduction

Machine learning models based on deep neural networks have achieved unprecedented performance across a wide range of tasks [1, 2, 3]. Typically, these models are regarded as complex systems for which many types of theoretical analyses are intractable. Moreover, characterizing the gradient-based training dynamics of these models is challenging owing to the typically high-dimensional non-convex loss surfaces governing the optimization. As is common in the physical sciences, investigating the extreme limits of such systems can often shed light on these hard problems. For neural networks, one such limit is that of infinite width, which refers either to the number of hidden units in a fully-connected layer or to the number of channels in a convolutional layer. Under this limit, the output of the network at initialization is a draw from a Gaussian process (GP); moreover, the network output remains governed by a GP after exact Bayesian training using squared loss [4, 5, 6, 7, 8]. Aside from its theoretical simplicity, the infinite-width limit is also of practical interest as wider networks have been found to generalize better [5, 7, 9, 10, 11].

In this work, we explore the learning dynamics of wide neural networks under gradient descent and find that the weight-space description of the dynamics becomes surprisingly simple: as the width becomes large, the neural network can be effectively replaced by its first-order Taylor expansion with respect to its parameters at initialization. For this linear model, the dynamics of gradient descent become *analytically tractable*. While the linearization is only exact in the infinite width limit, we nevertheless find excellent agreement between the predictions of the original network and those of

---

[*]Both authors contributed equally to this work. Work done as a member of the Google AI Residency program (https://g.co/airesidency).

the linearized version even for finite width configurations. The agreement persists across different architectures, optimization methods, and loss functions.

For squared loss, the exact learning dynamics admit a closed-form solution that allows us to characterize the evolution of the predictive distribution in terms of a GP. This result can be thought of as an extension of "sample-then-optimize" posterior sampling [12] to the training of deep neural networks. Our empirical simulations confirm that the result accurately models the variation in predictions across an ensemble of finite-width models with different random initializations.

Here we summarize our contributions:

- **Parameter space dynamics**: We show that wide network training dynamics in parameter space are equivalent to the training dynamics of a model which is affine in the collection of all network parameters, the weights and biases. This result holds regardless of the choice of loss function. For squared loss, the dynamics admit a closed-form solution as a function of time.

- **Sufficient conditions for linearization**: We formally prove that there exists a threshold learning rate $\eta_{\text{critical}}$ (see Theorem 2.1), such that gradient descent training trajectories with learning rate smaller than $\eta_{\text{critical}}$ stay in an $\mathcal{O}\left(n^{-1/2}\right)$-neighborhood of the trajectory of the linearized network when $n$, the width of the hidden layers, is sufficiently large.

- **Output distribution dynamics**: We formally show that the predictions of a neural network throughout gradient descent training are described by a GP as the width goes to infinity (see Theorem 2.2), extending results from Jacot et al. [13]. We further derive explicit time-dependent expressions for the evolution of this GP during training. Finally, we provide a novel interpretation of the result. In particular, it offers a quantitative understanding of the mechanism by which gradient descent differs from Bayesian posterior sampling of the parameters: while both methods generate draws from a GP, gradient descent does not generate samples from the posterior of any probabilistic model.

- **Large scale experimental support**: We empirically investigate the applicability of the theory in the finite-width setting and find that it gives an accurate characterization of both learning dynamics and posterior function distributions across a variety of conditions, including some practical network architectures such as the wide residual network [14].

- **Parameterization independence**: We note that linearization result holds both in standard and NTK parameterization (defined in §2.1), while previous work assumed the latter, emphasizing that the effect is due to increase in width rather than the particular parameterization.

- **Analytic** $\mathrm{ReLU}$ **and** $\mathrm{erf}$ **neural tangent kernels**: We compute the analytic neural tangent kernel corresponding to fully-connected networks with $\mathrm{ReLU}$ or $\mathrm{erf}$ nonlinearities.

- **Source code**: Example code investigating both function space and parameter space linearized learning dynamics described in this work is released as open source code within [15].[2] We also provide accompanying interactive Colab notebooks for both **parameter space**[3] and **function space**[4] linearization.

## 1.1 Related work

We build on recent work by Jacot et al. [13] that characterize the exact dynamics of network outputs throughout gradient descent training in the infinite width limit. Their results establish that full batch gradient descent in parameter space corresponds to kernel gradient descent in function space with respect to a new kernel, the Neural Tangent Kernel (NTK). We examine what this implies about dynamics in parameter space, where training updates are actually made.

Daniely et al. [16] study the relationship between neural networks and kernels at initialization. They bound the difference between the infinite width kernel and the empirical kernel at finite width $n$, which diminishes as $\mathcal{O}(1/\sqrt{n})$. Daniely [17] uses the same kernel perspective to study stochastic gradient descent (SGD) training of neural networks.

Saxe et al. [18] study the training dynamics of deep linear networks, in which the nonlinearities are treated as identity functions. Deep linear networks are linear in their inputs, but not in their

parameters. In contrast, we show that the outputs of sufficiently wide neural networks are linear in the updates to their parameters during gradient descent, but not usually their inputs.

Du et al. [19], Allen-Zhu et al. [20, 21], Zou et al. [22] study the convergence of gradient descent to global minima. They proved that for i.i.d. Gaussian initialization, the parameters of sufficiently wide networks move little from their initial values during SGD. This small motion of the parameters is crucial to the effect we present, where wide neural networks behave linearly in terms of their parameters throughout training.

Mei et al. [23], Chizat and Bach [24], Rotskoff and Vanden-Eijnden [25], Sirignano and Spiliopoulos [26] analyze the mean field SGD dynamics of training neural networks in the large-width limit. Their mean field analysis describes distributional dynamics of network parameters via a PDE. However, their analysis is restricted to one hidden layer networks with a scaling limit $(1/n)$ different from ours $(1/\sqrt{n})$, which is commonly used in modern networks [2, 27].

Chizat et al. [28][5] argued that infinite width networks are in 'lazy training' regime and maybe too simple to be applicable to realistic neural networks. Nonetheless, we empirically investigate the applicability of the theory in the finite-width setting and find that it gives an accurate characterization of both the learning dynamics and posterior function distributions across a variety of conditions, including some practical network architectures such as the wide residual network [14].

## 2 Theoretical results

### 2.1 Notation and setup for architecture and training dynamics

Let $\mathcal{D} \subseteq \mathbb{R}^{n_0} \times \mathbb{R}^k$ denote the training set and $\mathcal{X} = \{x : (x, y) \in \mathcal{D}\}$ and $\mathcal{Y} = \{y : (x, y) \in \mathcal{D}\}$ denote the inputs and labels, respectively. Consider a fully-connected feed-forward network with $L$ hidden layers with widths $n_l$, for $l = 1, ..., L$ and a readout layer with $n_{L+1} = k$. For each $x \in \mathbb{R}^{n_0}$, we use $h^l(x), x^l(x) \in \mathbb{R}^{n_l}$ to represent the pre- and post-activation functions at layer $l$ with input $x$. The recurrence relation for a feed-forward network is defined as

$$\begin{cases} h^{l+1} & = x^l W^{l+1} + b^{l+1} \\ x^{l+1} & = \phi\left(h^{l+1}\right) \end{cases} \text{ and } \begin{cases} W^l_{i,j} & = \frac{\sigma_\omega}{\sqrt{n_l}} \omega^l_{ij} \\ b^l_j & = \sigma_b \beta^l_j \end{cases}, \tag{1}$$

where $\phi$ is a point-wise activation function, $W^{l+1} \in \mathbb{R}^{n_l \times n_{l+1}}$ and $b^{l+1} \in \mathbb{R}^{n_{l+1}}$ are the weights and biases, $\omega^l_{ij}$ and $b^l_j$ are the trainable variables, drawn i.i.d. from a standard Gaussian $\omega^l_{ij}, \beta^l_j \sim \mathcal{N}(0, 1)$ at initialization, and $\sigma^2_\omega$ and $\sigma^2_b$ are weight and bias variances. Note that this parametrization is non-standard, and we will refer to it as the NTK parameterization. It has already been adopted in several recent works [29, 30, 13, 19, 31]. Unlike the standard parameterization that only normalizes the forward dynamics of the network, the NTK-parameterization also normalizes its backward dynamics. We note that the predictions and training dynamics of NTK-parameterized networks are identical to those of standard networks, up to a width-dependent scaling factor in the learning rate for each parameter tensor. As we derive, and support experimentally, in Supplementary Material (SM) §F and §G, our results (linearity in weights, GP predictions) also hold for networks with a standard parameterization.

We define $\theta^l \equiv \text{vec}\left(\{W^l, b^l\}\right)$, the $((n_{l-1}+1)n_l) \times 1$ vector of all parameters for layer $l$. $\theta = \text{vec}\left(\cup_{l=1}^{L+1} \theta^l\right)$ then indicates the vector of all network parameters, with similar definitions for $\theta^{\leq l}$ and $\theta^{>l}$. Denote by $\theta_t$ the time-dependence of the parameters and by $\theta_0$ their initial values. We use $f_t(x) \equiv h^{L+1}(x) \in \mathbb{R}^k$ to denote the output (or logits) of the neural network at time $t$. Let $\ell(\hat{y}, y) : \mathbb{R}^k \times \mathbb{R}^k \to \mathbb{R}$ denote the loss function where the first argument is the prediction and the second argument the true label. In supervised learning, one is interested in learning a $\theta$ that minimizes the empirical loss[6], $\mathcal{L} = \sum_{(x,y) \in \mathcal{D}} \ell(f_t(x, \theta), y)$.

Let $\eta$ be the learning rate[7]. Via continuous time gradient descent, the evolution of the parameters $\theta$ and the logits $f$ can be written as

$$\dot{\theta}_t = -\eta \nabla_\theta f_t(\mathcal{X})^T \nabla_{f_t(\mathcal{X})} \mathcal{L} \tag{2}$$

$$\dot{f}_t(\mathcal{X}) = \nabla_\theta f_t(\mathcal{X})\, \dot{\theta}_t = -\eta\, \hat{\Theta}_t(\mathcal{X}, \mathcal{X}) \nabla_{f_t(\mathcal{X})} \mathcal{L} \tag{3}$$

where $f_t(\mathcal{X}) = \text{vec}\left([f_t(x)]_{x \in \mathcal{X}}\right)$, the $k|\mathcal{D}| \times 1$ vector of concatenated logits for all examples, and $\nabla_{f_t(\mathcal{X})}\mathcal{L}$ is the gradient of the loss with respect to the model's output, $f_t(\mathcal{X})$. $\hat{\Theta}_t \equiv \hat{\Theta}_t(\mathcal{X}, \mathcal{X})$ is the tangent kernel at time $t$, which is a $k|\mathcal{D}| \times k|\mathcal{D}|$ matrix

$$\hat{\Theta}_t = \nabla_\theta f_t(\mathcal{X}) \nabla_\theta f_t(\mathcal{X})^T = \sum_{l=1}^{L+1} \nabla_{\theta^l} f_t(\mathcal{X}) \nabla_{\theta^l} f_t(\mathcal{X})^T . \tag{4}$$

One can define the tangent kernel for general arguments, e.g. $\hat{\Theta}_t(x, \mathcal{X})$ where $x$ is test input. At finite-width, $\hat{\Theta}$ will depend on the specific random draw of the parameters and in this context we refer to it as the *empirical* tangent kernel.

The dynamics of discrete gradient descent can be obtained by replacing $\dot{\theta}_t$ and $\dot{f}_t(\mathcal{X})$ with $(\theta_{i+1} - \theta_i)$ and $(f_{i+1}(\mathcal{X}) - f_i(\mathcal{X}))$ above, and replacing $e^{-\eta \hat{\Theta}_0 t}$ with $(1 - (1 - \eta\hat{\Theta}_0)^i)$ below.

## 2.2 Linearized networks have closed form training dynamics for parameters and outputs

In this section, we consider the training dynamics of the linearized network. Specifically, we replace the outputs of the neural network by their first order Taylor expansion,

$$f_t^{\text{lin}}(x) \equiv f_0(x) + \nabla_\theta f_0(x)|_{\theta=\theta_0}\, \omega_t , \tag{5}$$

where $\omega_t \equiv \theta_t - \theta_0$ is the change in the parameters from their initial values. Note that $f_t^{\text{lin}}$ is the sum of two terms: the first term is the initial output of the network, which remains unchanged during training, and the second term captures the change to the initial value during training. The dynamics of gradient flow using this linearized function are governed by,

$$\dot{\omega}_t = -\eta \nabla_\theta f_0(\mathcal{X})^T \nabla_{f_t^{\text{lin}}(\mathcal{X})} \mathcal{L} \tag{6}$$

$$\dot{f}_t^{\text{lin}}(x) = -\eta\, \hat{\Theta}_0(x, \mathcal{X}) \nabla_{f_t^{\text{lin}}(\mathcal{X})} \mathcal{L} . \tag{7}$$

As $\nabla_\theta f_0(x)$ remains constant throughout training, these dynamics are often quite simple. In the case of an MSE loss, i.e., $\ell(\hat{y}, y) = \frac{1}{2} \|\hat{y} - y\|_2^2$, the ODEs have closed form solutions

$$\omega_t = -\nabla_\theta f_0(\mathcal{X})^T \hat{\Theta}_0^{-1} \left(I - e^{-\eta\hat{\Theta}_0 t}\right) (f_0(\mathcal{X}) - \mathcal{Y}) , \tag{8}$$

$$f_t^{\text{lin}}(\mathcal{X}) = (I - e^{-\eta\hat{\Theta}_0 t})\mathcal{Y} + e^{-\eta\hat{\Theta}_0 t} f_0(\mathcal{X}) . \tag{9}$$

For an arbitrary point $x$, $f_t^{\text{lin}}(x) = \mu_t(x) + \gamma_t(x)$, where

$$\mu_t(x) = \hat{\Theta}_0(x, \mathcal{X})\hat{\Theta}_0^{-1} \left(I - e^{-\eta\hat{\Theta}_0 t}\right) \mathcal{Y} \tag{10}$$

$$\gamma_t(x) = f_0(x) - \hat{\Theta}_0(x, \mathcal{X})\hat{\Theta}_0^{-1} \left(I - e^{-\eta\hat{\Theta}_0 t}\right) f_0(\mathcal{X}) . \tag{11}$$

Therefore, we can obtain the time evolution of the linearized neural network without running gradient descent. We only need to compute the tangent kernel $\hat{\Theta}_0$ and the outputs $f_0$ at initialization and use Equations 8, 10, and 11 to compute the dynamics of the weights and the outputs.

## 2.3 Infinite width limit yields a Gaussian process

As the width of the hidden layers approaches infinity, the Central Limit Theorem (CLT) implies that the outputs at initialization $\{f_0(x)\}_{x \in \mathcal{X}}$ converge to a multivariate Gaussian in distribution.

Informally, this occurs because the pre-activations at each layer are a sum of Gaussian random variables (the weights and bias), and thus become a Gaussian random variable themselves. See [32, 33, 5, 34, 35] for more details, and [36, 7] for a formal treatment.

Therefore, randomly initialized neural networks are in correspondence with a certain class of GPs (hereinafter referred to as NNGPs), which facilitates a fully Bayesian treatment of neural networks [5, 6]. More precisely, let $f_t^i$ denote the $i$-th output dimension and $\mathcal{K}$ denote the sample-to-sample kernel function (of the pre-activation) of the outputs in the infinite width setting,

$$\mathcal{K}^{i,j}(x, x') = \lim_{\min(n_1, \ldots, n_L) \to \infty} \mathbb{E}\left[ f_0^i(x) \cdot f_0^j(x') \right], \tag{12}$$

then $f_0(\mathcal{X}) \sim \mathcal{N}(0, \mathcal{K}(\mathcal{X}, \mathcal{X}))$, where $\mathcal{K}^{i,j}(x, x')$ denotes the covariance between the $i$-th output of $x$ and $j$-th output of $x'$, which can be computed recursively (see Lee et al. [5, §2.3] and SM §E). For a test input $x \in \mathcal{X}_T$, the joint output distribution $f([x, \mathcal{X}])$ is also multivariate Gaussian. Conditioning on the training samples[8], $f(\mathcal{X}) = \mathcal{Y}$, the distribution of $f(x)|\mathcal{X}, \mathcal{Y}$ is also a Gaussian $\mathcal{N}(\mu(x), \Sigma(x))$,

$$\mu(x) = \mathcal{K}(x, \mathcal{X})\mathcal{K}^{-1}\mathcal{Y}, \quad \Sigma(x) = \mathcal{K}(x, x) - \mathcal{K}(x, \mathcal{X})\mathcal{K}^{-1}\mathcal{K}(x, \mathcal{X})^T, \tag{13}$$

and where $\mathcal{K} = \mathcal{K}(\mathcal{X}, \mathcal{X})$. This is the posterior predictive distribution resulting from exact Bayesian inference in an infinitely wide neural network.

### 2.3.1 Gaussian processes from gradient descent training

If we freeze the variables $\theta^{\leq L}$ after initialization and only optimize $\theta^{L+1}$, the original network and its linearization are identical. Letting the width approach infinity, this particular tangent kernel $\hat{\Theta}_0$ will converge to $\mathcal{K}$ in probability and Equation 10 will converge to the posterior Equation 13 as $t \to \infty$ (for further details see SM §D). This is a realization of the "sample-then-optimize" approach for evaluating the posterior of a Gaussian process proposed in Matthews et al. [12].

If none of the variables are frozen, in the infinite width setting, $\hat{\Theta}_0$ also converges in probability to a deterministic kernel $\Theta$ [13, 37], which we sometimes refer to as the analytic kernel, and which can also be computed recursively (see SM §E). For ReLU and erf nonlinearity, $\Theta$ can be exactly computed (SM §C) which we use in §3. Letting the width go to infinity, for any $t$, the output $f_t^{\text{lin}}(x)$ of the linearized network is also Gaussian distributed because Equations 10 and 11 describe an affine transform of the Gaussian $[f_0(x), f_0(\mathcal{X})]$. Therefore

**Corollary 1.** *For every test points in $x \in \mathcal{X}_T$, and $t \geq 0$, $f_t^{lin}(x)$ converges in distribution as width goes to infinity to a Gaussian with mean and covariance given by[9]*

$$\mu(\mathcal{X}_T) = \Theta(\mathcal{X}_T, \mathcal{X})\Theta^{-1}\left(I - e^{-\eta\Theta t}\right)\mathcal{Y}, \tag{14}$$

$$\Sigma(\mathcal{X}_T, \mathcal{X}_T) = \mathcal{K}(\mathcal{X}_T, \mathcal{X}_T) + \Theta(\mathcal{X}_T, \mathcal{X})\Theta^{-1}\left(I - e^{-\eta\Theta t}\right)\mathcal{K}\left(I - e^{-\eta\Theta t}\right)\Theta^{-1}\Theta(\mathcal{X}, \mathcal{X}_T)$$
$$- \left(\Theta(\mathcal{X}_T, \mathcal{X})\Theta^{-1}\left(I - e^{-\eta\Theta t}\right)\mathcal{K}(\mathcal{X}, \mathcal{X}_T) + h.c.\right). \tag{15}$$

*Therefore, over random initialization, $\lim_{t \to \infty} \lim_{n \to \infty} f_t^{lin}(x)$ has distribution*

$$\mathcal{N}\big(\Theta(\mathcal{X}_T, \mathcal{X})\Theta^{-1}\mathcal{Y},$$
$$\mathcal{K}(\mathcal{X}_T, \mathcal{X}_T) + \Theta(\mathcal{X}_T, \mathcal{X})\Theta^{-1}\mathcal{K}\Theta^{-1}\Theta(\mathcal{X}, \mathcal{X}_T) - \left(\Theta(\mathcal{X}_T, \mathcal{X})\Theta^{-1}\mathcal{K}(\mathcal{X}, \mathcal{X}_T) + h.c.\right)\big). \tag{16}$$

Unlike the case when only $\theta^{L+1}$ is optimized, Equations 14 and 15 do not admit an interpretation corresponding to the posterior sampling of a probabilistic model.[10] We contrast the predictive distributions from the NNGP, NTK-GP (i.e. Equations 14 and 15) and ensembles of NNs in Figure 2.

Infinitely-wide neural networks open up ways to study deep neural networks both under fully Bayesian training through the Gaussian process correspondence, and under GD training through the linearization perspective. The resulting distributions over functions are inconsistent (the distribution resulting

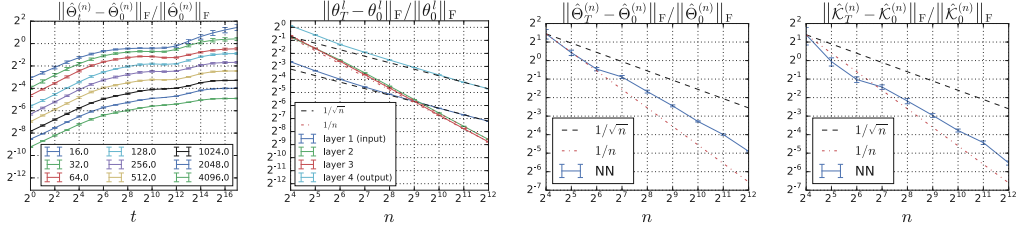

Figure 1: **Relative Frobenius norm change during training.** Three hidden layer ReLU networks trained with $\eta = 1.0$ on a subset of MNIST ($|\mathcal{D}| = 128$). We measure changes of (input/output/intermediary) weights, empirical $\hat{\Theta}$, and empirical $\hat{\mathcal{K}}$ after $T = 2^{17}$ steps of gradient descent updates for varying width. We see that the relative change in input/output weights scales as $1/\sqrt{n}$ while intermediate weights scales as $1/n$, this is because the dimension of the input/output does not grow with $n$. The change in $\hat{\Theta}$ and $\hat{\mathcal{K}}$ is upper bounded by $\mathcal{O}\left(1/\sqrt{n}\right)$ but is closer to $\mathcal{O}\left(1/n\right)$. See Figure S6 for the same experiment with 3-layer $\tanh$ and 1-layer ReLU networks. See Figures S9 and S10 for additional comparisons of finite width empirical and analytic kernels.

from GD training does not generally correspond to a Bayesian posterior). We believe understanding the biases over learned functions induced by different training schemes and architectures is a fascinating avenue for future work.

## 2.4   Infinite width networks are linearized networks

Equation 2 and 3 of the original network are intractable in general, since $\hat{\Theta}_t$ evolves with time. However, for the mean squared loss, we are able to prove formally that, as long as the learning rate $\eta < \eta_{\text{critical}} := 2(\lambda_{\min}(\Theta) + \lambda_{\max}(\Theta))^{-1}$, where $\lambda_{\min/\max}(\Theta)$ is the min/max eigenvalue of $\Theta$, the gradient descent dynamics of the original neural network falls into its linearized dynamics regime.

**Theorem 2.1** (Informal). *Let $n_1 = \cdots = n_L = n$ and assume $\lambda_{\min}(\Theta) > 0$. Applying gradient descent with learning rate $\eta < \eta_{\text{critical}}$ (or gradient flow), for every $x \in \mathbb{R}^{n_0}$ with $\|x\|_2 \leq 1$, with probability arbitrarily close to 1 over random initialization,*

$$\sup_{t \geq 0} \left\| f_t(x) - f_t^{lin}(x) \right\|_2, \ \sup_{t \geq 0} \frac{\|\theta_t - \theta_0\|_2}{\sqrt{n}}, \ \sup_{t \geq 0} \left\| \hat{\Theta}_t - \hat{\Theta}_0 \right\|_F = \mathcal{O}(n^{-\frac{1}{2}}), \ \text{as} \quad n \to \infty. \quad (17)$$

Therefore, as $n \to \infty$, the distributions of $f_t(x)$ and $f_t^{\text{lin}}(x)$ become the same. Coupling with Corollary 1, we have

**Theorem 2.2.** *If $\eta < \eta_{\text{critical}}$, then for every $x \in \mathbb{R}^{n_0}$ with $\|x\|_2 \leq 1$, as $n \to \infty$, $f_t(x)$ converges in distribution to the Gaussian with mean and variance given by Equation 14 and Equation 15.*

We refer the readers to Figure 2 for empirical verification of this theorem. The proof of Theorem 2.1 consists of two steps. The first step is to prove the global convergence of overparameterized neural networks [19, 20, 21, 22] and stability of the NTK under gradient descent (and gradient flow); see SM §G. This stability was first observed and proved in [13] in the gradient flow and sequential limit (i.e. letting $n_1 \to \infty, \ldots, n_L \to \infty$ sequentially) setting under certain assumptions about global convergence of gradient flow. In §G, we show how to use the NTK to provide a self-contained (and cleaner) proof of such global convergence and the stability of NTK simultaneously. The second step is to couple the stability of NTK with Grönwall's type arguments [38] to upper bound the discrepancy between $f_t$ and $f_t^{\text{lin}}$, i.e. the first norm in Equation 17. Intuitively, the ODE of the original network (Equation 3) can be considered as a $\|\hat{\Theta}_t - \hat{\Theta}_0\|_F$-fluctuation from the linearized ODE (Equation 7). One expects the difference between the solutions of these two ODEs to be upper bounded by some functional of $\|\hat{\Theta}_t - \hat{\Theta}_0\|_F$; see SM §H. Therefore, for a large width network, the training dynamics can be well approximated by linearized dynamics.

Note that the updates for individual weights in Equation 6 vanish in the infinite width limit, which for instance can be seen from the explicit width dependence of the gradients in the NTK parameterization. Individual weights move by a vanishingly small amount for wide networks in this regime of dynamics, as do hidden layer activations, but they collectively conspire to provide a finite change in the final output of the network, as is necessary for training. An additional insight gained from linearization

of the network is that the individual instance dynamics derived in [13] can be viewed as a random features method,[11] where the features are the gradients of the model with respect to its weights.

## 2.5 Extensions to other optimizers, architectures, and losses

Our theoretical analysis thus far has focused on fully-connected single-output architectures trained by full batch gradient descent. In SM §B we derive corresponding results for: networks with multi-dimensional outputs, training against a cross entropy loss, and gradient descent with momentum.

In addition to these generalizations, there is good reason to suspect the results to extend to much broader class of models and optimization procedures. In particular, a wealth of recent literature suggests that the mean field theory governing the wide network limit of fully-connected models [32, 33] extends naturally to residual networks [35], CNNs [34], RNNs [39], batch normalization [40], and to broad architectures [37]. We postpone the development of these additional theoretical extensions in favor of an empirical investigation of linearization for a variety of architectures.

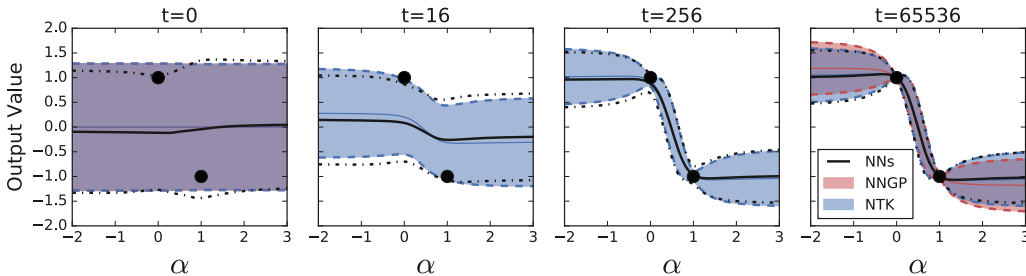

Figure 2: **Dynamics of mean and variance of trained neural network outputs follow analytic dynamics from linearization**. Black lines indicate the time evolution of the predictive output distribution from an ensemble of 100 trained neural networks (NNs). The blue region indicates the analytic prediction of the output distribution throughout training (Equations 14, 15). Finally, the red region indicates the prediction that would result from training only the top layer, corresponding to an NNGP (Equations S22, S23). The trained network has 3 hidden layers of width 8192, $\tanh$ activation functions, $\sigma_w^2 = 1.5$, no bias, and $\eta = 0.5$. The output is computed for inputs interpolated between two training points (denoted with black dots) $x(\alpha) = \alpha x^{(1)} + (1 - \alpha)x^{(2)}$. The shaded region and dotted lines denote 2 standard deviations ($\sim 95\%$ quantile) from the mean denoted in solid lines. Training was performed with full-batch gradient descent with dataset size $|\mathcal{D}| = 128$. For dynamics for individual function initializations, see SM Figure S1.

## 3 Experiments

In this section, we provide empirical support showing that the training dynamics of wide neural networks are well captured by linearized models. We consider fully-connected, convolutional, and wide ResNet architectures trained with full- and mini- batch gradient descent using learning rates sufficiently small so that the continuous time approximation holds well. We consider two-class classification on CIFAR-10 (horses and planes) as well as ten-class classification on MNIST and CIFAR-10. When using MSE loss, we treat the binary classification task as regression with one class regressing to $+1$ and the other to $-1$.

Experiments in Figures 1, 4, S2, S3, S4, S5 and S6, were done in JAX [41]. The remaining experiments used TensorFlow [42]. An open source implementation of this work providing tools to investigate linearized learning dynamics is available at `www.github.com/google/neural-tangents` [15].

**Predictive output distribution**: In the case of an MSE loss, the output distribution remains Gaussian throughout training. In Figure 2, the predictive output distribution for input points interpolated between two training points is shown for an ensemble of neural networks and their corresponding GPs. The interpolation is given by $x(\alpha) = \alpha x^{(1)} + (1 - \alpha)x^{(2)}$ where $x^{(1,2)}$ are two training inputs

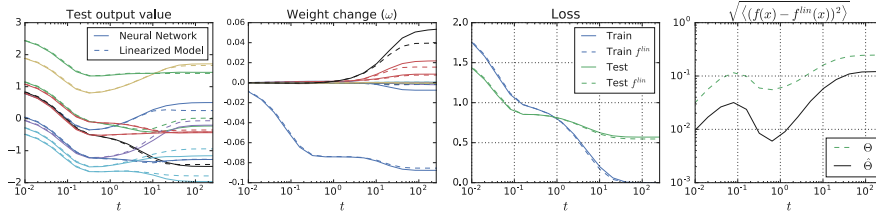

Figure 3: **Full batch gradient descent on a model behaves similarly to analytic dynamics on its linearization, both for network outputs, and also for individual weights.** A binary CIFAR classification task with MSE loss and a $\mathrm{ReLU}$ fully-connected network with 5 hidden layers of width $n = 2048$, $\eta = 0.01$, $|\mathcal{D}| = 256$, $k = 1$, $\sigma_w^2 = 2.0$, and $\sigma_b^2 = 0.1$. Left two panes show dynamics for a randomly selected subset of datapoints or parameters. Third pane shows that the dynamics of loss for training and test points agree well between the original and linearized model. The last pane shows the dynamics of RMSE between the two models on test points. We observe that the empirical kernel $\hat{\Theta}$ gives more accurate dynamics for finite width networks.

with different classes. We observe that the mean and variance dynamics of neural network outputs during gradient descent training follow the analytic dynamics from linearization well (Equations 14, 15). Moreover the NNGP predictive distribution which corresponds to exact Bayesian inference, while similar, is noticeably *different* from the predictive distribution at the end of gradient descent training. For dynamics for individual function draws see SM Figure S1.

**Comparison of training dynamics of linearized network to original network**: For a particular realization of a finite width network, one can analytically predict the dynamics of the weights and outputs over the course of training using the empirical tangent kernel at initialization. In Figures 3, 4 (see also S2, S3), we compare these linearized dynamics (Equations 8, 9) with the result of training the actual network. In all cases we see remarkably good agreement. We also observe that for finite networks, dynamics predicted using the empirical kernel $\hat{\Theta}$ better match the data than those obtained using the infinite-width, analytic, kernel $\Theta$. To understand this we note that $\|\hat{\Theta}_T^{(n)} - \hat{\Theta}_0^{(n)}\|_F = \mathcal{O}(1/n) \leq \mathcal{O}(1/\sqrt{n}) = \|\hat{\Theta}_0^{(n)} - \Theta\|_F$, where $\hat{\Theta}_0^{(n)}$ denotes the empirical tangent kernel of width $n$ network, as plotted in Figure 1.

One can directly optimize parameters of $f^{\mathrm{lin}}$ instead of solving the ODE induced by the tangent kernel $\hat{\Theta}$. Standard neural network optimization techniques such as mini-batching, weight decay, and data augmentation can be directly applied. In Figure 4 (S2, S3), we compared the training dynamics of the linearized and original network while directly training both networks.

With direct optimization of linearized model, we tested full ($|\mathcal{D}| = 50,000$) MNIST digit classification with cross-entropy loss, and trained with a momentum optimizer (Figure S3). For cross-entropy loss with softmax output, some logits at late times grow indefinitely, in contrast to MSE loss where logits converge to target value. The error between original and linearized model for cross entropy loss becomes much worse at late times if the two models deviate significantly before the logits enter their late-time steady-growth regime (See Figure S4).

Linearized dynamics successfully describes the training of networks beyond vanilla fully-connected models. To demonstrate the generality of this procedure we show we can predict the learning dynamics of subclass of Wide Residual Networks (WRNs) [14]. WRNs are a class of model that are popular in computer vision and leverage convolutions, batch normalization, skip connections, and average pooling. In Figure 4, we show a comparison between the linearized dynamics and the true dynamics for a wide residual network trained with MSE loss and SGD with momentum, *trained on the full CIFAR-10 dataset*. We slightly modified the block structure described in Table S1 so that each layer has a constant number of channels (1024 in this case), and otherwise followed the original implementation. As elsewhere, we see strong agreement between the predicted dynamics and the result of training.

**Effects of dataset size**: The training dynamics of a neural network match those of its linearization when the width is infinite and the dataset is finite. In previous experiments, we chose sufficiently wide networks to achieve small error between neural networks and their linearization for smaller

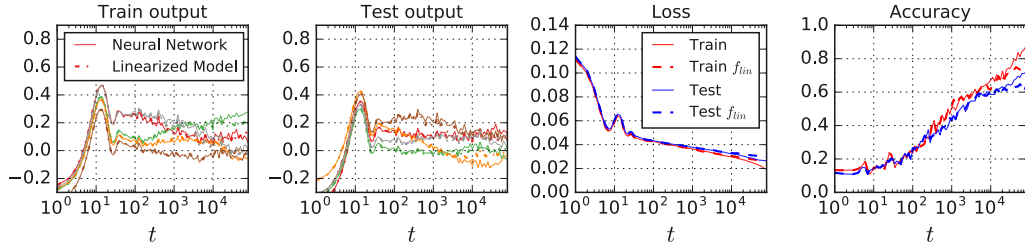

Figure 4: **A wide residual network and its linearization behave similarly when both are trained by SGD with momentum on MSE loss on CIFAR-10.** We adopt the network architecture from Zagoruyko and Komodakis [14]. We use $N = 1$, channel size 1024, $\eta = 1.0$, $\beta = 0.9$, $k = 10$, $\sigma_w^2 = 1.0$, and $\sigma_b^2 = 0.0$. See Table S1 for details of the architecture. Both the linearized and original model are trained directly on full CIFAR-10 ($|\mathcal{D}| = 50,000$), using SGD with batch size 8. Output dynamics for a randomly selected subset of train and test points are shown in the first two panes. Last two panes show training and accuracy curves for the original and linearized networks.

datasets. Overall, we observe that as the width grows the error decreases (Figure S5). Additionally, we see that the error grows in the size of the dataset. Thus, although error grows with dataset this can be counterbalanced by a corresponding increase in the model size.

## 4    Discussion

We showed theoretically that the learning dynamics in parameter space of deep nonlinear neural networks are exactly described by a linearized model in the infinite width limit. Empirical investigation revealed that this agrees well with actual training dynamics and predictive distributions across fully-connected, convolutional, and even wide residual network architectures, as well as with different optimizers (gradient descent, momentum, mini-batching) and loss functions (MSE, cross-entropy). Our results suggest that a surprising number of realistic neural networks may be operating in the regime we studied. This is further consistent with recent experimental work showing that neural networks are often robust to re-initialization but not re-randomization of layers (Zhang et al. [43]).

In the regime we study, since the learning dynamics are fully captured by the kernel $\hat{\Theta}$ and the target signal, studying the properties of $\hat{\Theta}$ to determine trainability and generalization are interesting future directions. Furthermore, the infinite width limit gives us a simple characterization of both gradient descent and Bayesian inference. By studying properties of the NNGP kernel $\mathcal{K}$ and the tangent kernel $\Theta$, we may shed light on the inductive bias of gradient descent.

Some layers of modern neural networks may be operating far from the linearized regime. Preliminary observations in Lee et al. [5] showed that wide neural networks trained with SGD perform similarly to the corresponding GPs as width increase, while GPs still outperform trained neural networks for both small and large dataset size. Furthermore, in Novak et al. [7], it is shown that the comparison of performance between finite- and infinite-width networks is highly architecture-dependent. In particular, it was found that infinite-width networks perform as well as or better than their finite-width counterparts for many fully-connected or locally-connected architectures. However, the opposite was found in the case of convolutional networks without pooling. It is still an open research question to determine the main factors that determine these performance gaps. We believe that examining the behavior of infinitely wide networks provides a strong basis from which to build up a systematic understanding of finite-width networks (and/or networks trained with large learning rates).

## Acknowledgements

We thank Greg Yang and Alex Alemi for useful discussions and feedback. We are grateful to Daniel Freeman, Alex Irpan and anonymous reviewers for providing valuable feedbacks on the draft. We thank the JAX team for developing a language which makes model linearization and NTK computation straightforward. We would like to especially thank Matthew Johnson for support and debugging help.

## Footnotes

[2]Note that the open source library has been expanded since initial submission of this work.

[3]colab.sandbox.google.com/github/google/neural-tangents/blob/master/notebooks/weight_space_linearization.ipynb

[4]colab.sandbox.google.com/github/google/neural-tangents/blob/master/notebooks/function_space_linearization.ipynb

[5]We note that this is a concurrent work and an expanded version of this note is presented in parallel at NeurIPS 2019.

[6]To simplify the notation for later equations, we use the *total* loss here instead of the *average* loss, but for all plots in §3, we show the *average* loss.

[7]Note that compared to the conventional parameterization, $\eta$ is larger by factor of width [31]. The NTK parameterization allows usage of a universal learning rate scale irrespective of network width.

[8] This imposes that $h^{L+1}$ directly corresponds to the network predictions. In the case of softmax readout, variational or sampling methods are required to marginalize over $h^{L+1}$.

[9] Here "+*h.c.*" is an abbreviation for "plus the Hermitian conjugate".

[10] One possible exception is when the NNGP kernel and NTK are the same up to a scalar multiplication. This is the case when the activation function is the identity function and there is no bias term.

[11]We thank Alex Alemi for pointing out a subtlety on correspondence to a random features method.

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
