[Supplementary Material]

# Supplementary Material

## A   Additional figures

Figure S1: **Sample of neural network outputs.** The lines correspond to the functions learned for 100 different initializations. The configuration is the same as in Figure 2.

Figure S2: **A convolutional network and its linearization behave similarly when trained using full batch gradient descent with a momentum optimizer.** Binary CIFAR classification task with MSE loss, $\tanh$ convolutional network with 3 hidden layers of channel size $n = 512$, $3 \times 3$ size filters, average pooling after last convolutional layer, $\eta = 0.1$, $\beta = 0.9$, $|\mathcal{D}| = 128$, $\sigma_w^2 = 2.0$ and $\sigma_b^2 = 0.1$. The linearized model is trained directly by full batch gradient descent with momentum, rather than by integrating its continuous time analytic dynamics. Panes are the same as in Figure 3.

Figure S3: **A neural network and its linearization behave similarly when both are trained via SGD with momentum on cross entropy loss on MNIST.** Experiment is for 10 class MNIST classification using a ReLU fully connected network with 2 hidden layers of width $n = 2048$, $\eta = 1.0$, $\beta = 0.9$, $|\mathcal{D}| = 50,000$, $k = 10$, $\sigma_w^2 = 2.0$, and $\sigma_b^2 = 0.1$. Both models are trained using stochastic minibatching with batch size 64. Panes are the same as in Figure 3, except that the top row shows all ten logits for a single randomly selected datapoint.

Figure S4: **Logit deviation for cross entropy loss.** Logits for models trained with cross entropy loss diverge at late times. If the deviation between the logits of the linearized model and original model are large early in training, as shown for the narrower networks (first row), logit deviation at late times can be significantly large. As the network becomes wider (second row), the logit deviates at a later point in training. Fully connected tanh network $L = 4$ trained on binary CIFAR classification problem.

Figure S5: **Error dependence on depth, width, and dataset size.** Final value of the RMSE for fully-connected, convolutional, wide residual network as networks become wider for varying depth and dataset size. Error in fully connected networks as the depth is varied from 1 to 16 (first) and the dataset size is varied from 32 to 4096 (last). Error in convolutional networks as the depth is varied between 1 and 32 (second), and WRN for depths 10 and 16 corresponding to N=1,2 described in Table S1 (third). Networks are critically initialized $\sigma_w^2 = 2.0$, $\sigma_b^2 = 0.1$, trained with gradient descent on MSE loss. Experiments in the first three panes used $|\mathcal{D}| = 128$.

Figure S6: **Relative Frobenius norm change during training.** *(top)* One hidden layer, ReLU networks trained with $\eta = 1.0$, on a 2-class CIFAR10 subset of size $|\mathcal{D}| = 128$. We measure changes of (read-out/non read-out) weights, empirical $\hat{\Theta}$ and empirical $\hat{\mathcal{K}}$ after $T = 2^{16}$ steps of gradient descent updates for varying width. *(bottom)* Networks with three layer $\tanh$ nonlinearity and other details are identical to Figure 1.

# B    Extensions

## B.1    Momentum

One direction is to go beyond vanilla gradient descent dynamics. We consider momentum updates[12]

$$\theta_{i+1} = \theta_i + \beta(\theta_i - \theta_{i-1}) - \eta\nabla_\theta\mathcal{L}|_{\theta=\theta_i} . \tag{S1}$$

The discrete update to the function output becomes

$$f_{i+1}^{\text{lin}}(x) = f_i^{\text{lin}}(x) - \eta\hat{\Theta}_0(x,\mathcal{X})\nabla_{f_i^{\text{lin}}(\mathcal{X})}\mathcal{L} + \beta(f_i^{\text{lin}}(x) - f_{i-1}^{\text{lin}}(x)) \tag{S2}$$

where $f_t^{\text{lin}}(x)$ is the output of the linearized network after $t$ steps. One can take the continuous time limit as in Qian [44], Su et al. [45] and obtain

$$\ddot{\omega}_t = \tilde{\beta}\dot{\omega}_t - \nabla_\theta f_0^{\text{lin}}(\mathcal{X})^T \nabla_{f_t^{\text{lin}}(\mathcal{X})}\mathcal{L} \tag{S3}$$

$$\ddot{f}_t^{\text{lin}}(x) = \tilde{\beta}\dot{f}_t^{\text{lin}}(x) - \hat{\Theta}_0(x,\mathcal{X})\nabla_{f_t^{\text{lin}}(\mathcal{X})}\mathcal{L} \tag{S4}$$

where continuous time relates to steps $t = i\sqrt{\eta}$ and $\tilde{\beta} = (\beta - 1)/\sqrt{\eta}$. These equations are also amenable to analytic treatment for MSE loss. See Figure S2, S3 and 4 for experimental agreement.

### B.2  Multi-dimensional output and cross-entropy loss

One can extend the loss function to general functions with multiple output dimensions. Unlike for squared error, we do not have a closed form solution to the dynamics equation. However, the equations for the dynamics can be solved using an ODE solver as an initial value problem.

$$\ell(f,y) = -\sum_i y^i \log \sigma(f^i), \qquad \sigma(f^i) \equiv \frac{\exp(f^i)}{\sum_j \exp(f^j)}. \tag{S5}$$

Recall that $\frac{\partial \ell}{\partial \hat{y}^i} = \sigma(\hat{y}^i) - y^i$. For general input point $x$ and for an arbitrary parameterized function $f^i(x)$ parameterized by $\theta$, gradient flow dynamics is given by

$$\dot{f}_t^i(x) = \nabla_\theta f_t^i(x)\frac{d\theta}{dt} = -\eta\nabla_\theta f_t^i(x)\sum_j \sum_{(z,y)\in\mathcal{D}} \left[\nabla_\theta f_t^j(z)^T \frac{\partial \ell(f_t, y)}{\partial \hat{y}^j}\right] \tag{S6}$$

$$= -\eta \sum_{(z,y)\in\mathcal{D}} \sum_j \nabla_\theta f_t^i(x)\nabla_\theta f_t^j(z)^T \left(\sigma(f_t^j(z)) - y^j\right) \tag{S7}$$

Let $\hat{\Theta}^{ij}(x,\mathcal{X}) = \nabla_\theta f^i(x)\nabla_\theta f^j(\mathcal{X})^T$. The above is

$$\dot{f}_t(\mathcal{X}) = -\eta\hat{\Theta}_t(\mathcal{X},\mathcal{X})\left(\sigma(f_t(\mathcal{X})) - \mathcal{Y}\right) \tag{S8}$$

$$\dot{f}_t(x) = -\eta\hat{\Theta}_t(x,\mathcal{X})\left(\sigma(f_t(\mathcal{X})) - \mathcal{Y}\right). \tag{S9}$$

The linearization is

$$\dot{f}_t^{\text{lin}}(\mathcal{X}) = -\eta\hat{\Theta}_0(\mathcal{X},\mathcal{X})\left(\sigma(f_t^{\text{lin}}(\mathcal{X})) - \mathcal{Y}\right) \tag{S10}$$

$$\dot{f}_t^{\text{lin}}(x) = -\eta\hat{\Theta}_0(x,\mathcal{X})\left(\sigma(f_t^{\text{lin}}(\mathcal{X})) - \mathcal{Y}\right). \tag{S11}$$

For general loss, e.g. cross-entropy with softmax output, we need to rely on solving the ODE Equations S10 and S11. We use the `dopri5` method for ODE integration, which is the default integrator in TensorFlow (`tf.contrib.integrate.odeint`).

## C  Neural Tangent kernel for ReLU and erf

For ReLU and erf activation functions, the tangent kernel can be computed analytically. We begin with the case $\phi = \text{ReLU}$; using the formula from Cho and Saul [46], we can compute $\mathcal{T}$ and $\dot{\mathcal{T}}$ in closed form. Let $\Sigma$ be a $2 \times 2$ PSD matrix. We will use

$$k_n(x,y) = \int \phi^n(x\cdot w)\phi^n(y\cdot w)e^{-\|w\|^2/2}dw \cdot (2\pi)^{-d/2} = \frac{1}{2\pi}\|x\|^n\|y\|^n J_n(\theta) \tag{S12}$$

where

$$\phi(x) = \max(x,0), \quad \theta(x,y) = \arccos\left(\frac{x\cdot y}{\|x\|\|y\|}\right),$$

$$J_0(\theta) = \pi - \theta, \quad J_1(\theta) = \sin\theta + (\pi - \theta)\cos\theta = \sqrt{1 - \left(\frac{x\cdot y}{\|x\|\|y\|}\right)^2} + (\pi - \theta)\left(\frac{x\cdot y}{\|x\|\|y\|}\right).$$
$$\tag{S13}$$

Let $d = 2$ and $u = (x \cdot w, y \cdot w)^T$. Then $u$ is a mean zero Gaussian with $\Sigma = [[x \cdot x, x \cdot y]; [x \cdot y, y \cdot y]]$. Then

$$\mathcal{T}(\Sigma) = k_1(x, y) = \frac{1}{2\pi} \|x\| \|y\| J_1(\theta) \tag{S14}$$

$$\dot{\mathcal{T}}(\Sigma) = k_0(x, y) = \frac{1}{2\pi} J_0(\theta) \tag{S15}$$

For $\phi = \mathrm{erf}$, let $\Sigma$ be the same as above. Following Williams [47], we get

$$\mathcal{T}(\Sigma) = \frac{2}{\pi} \sin^{-1} \left( \frac{2x \cdot y}{\sqrt{(1 + 2x \cdot x)(1 + 2y \cdot y)}} \right) \tag{S16}$$

$$\dot{\mathcal{T}}(\Sigma) = \frac{4}{\pi} \det(I + 2\Sigma)^{-1/2} \tag{S17}$$

## D  Gradient flow dynamics for training only the readout-layer

The connection between Gaussian processes and Bayesian wide neural networks can be extended to the setting when only the readout layer parameters are being optimized. More precisely, we show that when training only the readout layer, the outputs of the network form a Gaussian process (over an ensemble of draws from the parameter prior) throughout training, where that output is an interpolation between the GP prior and GP posterior.

Note that for any $x, x' \in \mathbb{R}^{n_0}$, in the infinite width limit $\bar{x}(x) \cdot \bar{x}(x') = \hat{\mathcal{K}}(x, x') \to \mathcal{K}(x, x')$ in probability, where for notational simplicity we assign $\bar{x}(x) = \left[ \frac{\sigma_w x^L(x)}{\sqrt{n_L}}, \sigma_b \right]$. The regression problem is specified with mean-squared loss

$$\mathcal{L} = \frac{1}{2} \|f(\mathcal{X}) - \mathcal{Y}\|_2^2 = \frac{1}{2} \|\bar{x}(\mathcal{X})\theta^{L+1} - \mathcal{Y}\|_2^2, \tag{S18}$$

and applying gradient flow to optimize the readout layer (and freezing all other parameters),

$$\dot{\theta}^{L+1} = -\eta \bar{x}(\mathcal{X})^T \left( \bar{x}(\mathcal{X})\theta^{L+1} - \mathcal{Y} \right), \tag{S19}$$

where $\eta$ is the learning rate. The solution to this ODE gives the evolution of the output of an arbitrary $x^*$. So long as the empirical kernel $\bar{x}(\mathcal{X})\bar{x}(\mathcal{X})^T$ is invertible, it is

$$f_t(x^*) = f_0(x^*) + \hat{\mathcal{K}}(x, \mathcal{X})\hat{\mathcal{K}}(\mathcal{X}, \mathcal{X})^{-1} \left( \exp\left( -\eta t \hat{\mathcal{K}}(\mathcal{X}, \mathcal{X}) \right) - I \right) (f_0(\mathcal{X}) - \mathcal{Y}) \tag{S20}$$

For any $x, x' \in \mathbb{R}^{n_0}$, letting $n_l \to \infty$ for $l = 1, \dots, L$, one has the convergence in distribution in probability and distribution respectively

$$\bar{x}(x)\bar{x}(x') \to \mathcal{K}(x, x') \quad \text{and} \quad \bar{x}(\mathcal{X})\theta_0^{L+1} \to \mathcal{N}(0, \mathcal{K}(\mathcal{X}, \mathcal{X})). \tag{S21}$$

Moreover $\bar{x}(\mathcal{X})\theta_0^{L+1}$ and the term containing $f_0(\mathcal{X})$ are the only stochastic term over the ensemble of network initializations, therefore for any $t$ the output $f(x^*)$ throughout training converges to a Gaussian distribution in the infinite width limit, with

$$\mathbb{E}[f_t(x^*)] = \mathcal{K}(x^*, \mathcal{X})\mathcal{K}^{-1}(I - e^{-\eta \mathcal{K} t})\mathcal{Y}, \tag{S22}$$

$$\mathrm{Var}[f_t(x^*)] = \mathcal{K}(x^*, x^*) - \mathcal{K}(x^*, \mathcal{X})\mathcal{K}^{-1}(I - e^{-2\eta \mathcal{K} t})\mathcal{K}(x^*, \mathcal{X})^T. \tag{S23}$$

Thus the output of the neural network is also a GP and the asymptotic solution (i.e. $t \to \infty$) is identical to the posterior of the NNGP (Equation 13). Therefore, in the infinite width case, the optimized neural network is performing posterior sampling if only the readout layer is being trained. This result is a realization of sample-then-optimize equivalence identified in Matthews et al. [12].

## E  Computing NTK and NNGP Kernel

For completeness, we reproduce, informally, the recursive formula of the NNGP kernel and the tangent kernel from [5] and [13], respectively. Let the activation function $\phi : \mathbb{R} \to \mathbb{R}$ be absolutely

continuous. Let $\mathcal{T}$ and $\dot{\mathcal{T}}$ be functions from $2 \times 2$ positive semi-definite matrices $\Sigma$ to $\mathbb{R}$ given by

$$\begin{cases} \mathcal{T}(\Sigma) = \mathbb{E}[\phi(u)\phi(v)] \\ \dot{\mathcal{T}}(\Sigma) = \mathbb{E}[\phi'(u)\phi'(v)] \end{cases} \qquad (u, v) \sim \mathcal{N}(0, \Sigma). \tag{S24}$$

In the infinite width limit, the NNGP and tangent kernel can be computed recursively. Let $x, x'$ be two inputs in $\mathbb{R}^{n_0}$. Then $h^l(x)$ and $h^l(x')$ converge in distribution to a joint Gaussian as $\min\{n_1, \ldots, n_{l-1}\}$. The mean is zero and the variance $\mathcal{K}^l(x, x')$ is

$$\mathcal{K}^l(x, x') = \tilde{\mathcal{K}}^l(x, x') \otimes \mathbf{Id}_{n_l} \tag{S25}$$

$$\tilde{\mathcal{K}}^l(x, x') = \sigma_\omega^2 \mathcal{T}\left(\begin{bmatrix} \tilde{\mathcal{K}}^{l-1}(x, x) & \tilde{\mathcal{K}}^{l-1}(x, x') \\ \tilde{\mathcal{K}}^{l-1}(x, x') & \tilde{\mathcal{K}}^{l-1}(x', x') \end{bmatrix}\right) + \sigma_b^2 \tag{S26}$$

with base case

$$\mathcal{K}^1(x, x') = \sigma_\omega^2 \cdot \frac{1}{n_0} x^T x' + \sigma_b^2. \tag{S27}$$

Using this one can also derive the tangent kernel for gradient descent training. We will use induction to show that

$$\Theta^l(x, x') = \tilde{\Theta}^l(x, x') \otimes \mathbf{Id}_{n_l} \tag{S28}$$

where

$$\tilde{\Theta}^l(x, x') = \tilde{\mathcal{K}}^l(x, x') + \sigma_\omega^2 \tilde{\Theta}^{l-1}(x, x') \dot{\mathcal{T}}\left(\begin{bmatrix} \tilde{\mathcal{K}}^{l-1}(x, x) & \tilde{\mathcal{K}}^{l-1}(x, x') \\ \tilde{\mathcal{K}}^{l-1}(x, x') & \tilde{\mathcal{K}}^{l-1}(x', x') \end{bmatrix}\right) \tag{S29}$$

with $\tilde{\Theta}^1 = \tilde{\mathcal{K}}^1$. Let

$$J^l(x) = \nabla_{\theta \le l} h_0^l(x) = [\nabla_{\theta^l} h_0^l(x), \nabla_{\theta < l} h_0^l(x)]. \tag{S30}$$

Then

$$J^l(x) J^l(x')^T = \nabla_{\theta^l} h_0^l(x) \nabla_{\theta^l} h_0^l(x')^T + \nabla_{\theta < l} h_0^l(x) \nabla_{\theta < l} h_0^l(x')^T \tag{S31}$$

Letting $n_1, \ldots, n_{l-1} \to \infty$ sequentially, the first term converges to the NNGP kernel $\mathcal{K}^l(x, x')$. By applying the chain rule and the induction step (letting $n_1, \ldots, n_{l-2} \to \infty$ sequentially), the second term is

$$\nabla_{\theta < l} h_0^l(x) \nabla_{\theta < l} h_0^l(x')^T = \frac{\partial h_0^l(x)}{\partial h_0^{l-1}(x)} \nabla_{\theta \le l-1} h_0^{l-1}(x) \nabla_{\theta \le l-1} h_0^{l-1}(x')^T \frac{\partial h_0^l(x')}{\partial h_0^{l-1}(x')}^T \tag{S32}$$

$$\to \frac{\partial h_0^l(x)}{\partial h_0^{l-1}(x)} \tilde{\Theta}^{l-1}(x, x') \otimes \mathbf{Id}_{n_{l-1}} \frac{\partial h_0^l(x')}{\partial h_0^{l-1}(x')}^T \qquad (n_1, \ldots, n_{l-2} \to \infty) \tag{S33}$$

$$\to \sigma_\omega^2 \left(\mathbb{E}\phi'(h_{0,i}^{l-1}(x)) \phi'(h_{0,i}^{l-1}(x')) \tilde{\Theta}^{l-1}(x, x')\right) \otimes \mathbf{Id}_{n_l} \qquad (n_{l-1} \to \infty) \tag{S34}$$

$$= \left(\sigma_\omega^2 \tilde{\Theta}^{l-1}(x, x') \dot{\mathcal{T}}\left(\begin{bmatrix} \tilde{\mathcal{K}}^{l-1}(x, x) & \tilde{\mathcal{K}}^{l-1}(x, x') \\ \tilde{\mathcal{K}}^{l-1}(x, x') & \tilde{\mathcal{K}}^{l-1}(x', x') \end{bmatrix}\right)\right) \otimes \mathbf{Id}_{n_l} \tag{S35}$$

## F    Results in function space for NTK parameterization transfer to standard parameterization

In this Section we present a sketch for why the function space linearization results, derived in [13] for NTK parameterized networks, also apply to networks with a standard parameterization. We follow this up with a formal proof in §G of the convergence of standard parameterization networks to their linearization in the limit of infinite width. A network with standard parameterization is described as:

$$\begin{cases} h^{l+1} &= x^l W^{l+1} + b^{l+1} \\ x^{l+1} &= \phi\left(h^{l+1}\right) \end{cases} \quad \text{and} \quad \begin{cases} W_{i,j}^l &= \omega_{ij}^l \sim \mathcal{N}\left(0, \frac{\sigma_\omega^2}{n_l}\right) \\ b_j^l &= \beta_j^l \sim \mathcal{N}\left(0, \sigma_b^2\right) \end{cases}. \tag{S36}$$

<table>
<tr><td>(a) MNIST</td><td>(b) CIFAR</td></tr>
</table>

Figure S7: **NTK vs Standard parameterization.** Across different choices of dataset, activation function and loss function, models obtained from (S)GD training for both parameterization (circle and triangle denotes NTK and standard parameterization respectively) get similar performance.

The NTK parameterization in Equation 1 is not commonly used for training neural networks. While the function that the network represents is the same for both NTK and standard parameterization, training dynamics under gradient descent are generally different for the two parameterizations. However, for a particular choice of layer-dependent learning rate training dynamics also become identical. Let $\eta^l_{\mathrm{NTK},w}$ and $\eta^l_{\mathrm{NTK},b}$ be layer-dependent learning rate for $W^l$ and $b^l$ in the NTK parameterization, and $\eta_{\mathrm{std}} = \frac{1}{n_{\max}}\eta_0$ be the learning rate for all parameters in the standard parameterization, where $n_{\max} = \max_l n_l$. Recall that gradient descent training in standard neural networks requires a learning rate that scales with width like $\frac{1}{n_{\max}}$, so $\eta_0$ defines a width-invariant learning rate [31]. If we choose

$$\eta^l_{\mathrm{NTK,\ w}} = \frac{n_l}{n_{\max}\sigma_\omega^2}\eta_0, \qquad \text{and} \qquad \eta^l_{\mathrm{NTK,\ b}} = \frac{1}{n_{\max}\sigma_b^2}\eta_0, \tag{S37}$$

then learning dynamics are identical for networks with NTK and standard parameterizations. With only extremely minor modifications, consisting of incorporating the multiplicative factors in Equation S37 into the per-layer contributions to the Jacobian, the arguments in §2.4 go through for an NTK network with learning rates defined in Equation S37. Since an NTK network with these learning rates exhibits identical training dynamics to a standard network with learning rate $\eta_{\mathrm{std}}$, the result in §2.4 that sufficiently wide NTK networks are linear in their parameters throughout training also applies to standard networks.

We can verify this property of networks with the standard parameterization experimentally. In Figure S7, we see that for different choices of dataset, activation function and loss function, final performance of two different parameterization leads to similar quality model for similar value of normalized learning rate $\eta_{\mathrm{std}} = \eta_{\mathrm{NTK}}/n$. Also, in Figure S8, we observe that our results is not due to the parameterization choice and holds for wide networks using the standard parameterization.

## G   Convergence of neural network to its linearization, and stability of NTK under gradient descent

In this section, we show that how to use the NTK to provide a simple proof of the global convergence of a neural network under (full-batch) gradient descent and the stability of NTK under gradient descent. We present the proof for standard parameterization. With some minor changes, the proof can also apply to the NTK parameterization. To lighten the notation, we only consider the asymptotic bound here. The neural networks are parameterized as in Equation S36. We make the following assumptions:

**Assumptions [1-4]:**

1. The widths of the hidden layers are identical, i.e. $n_1 = \cdots = n_L = n$ (our proof extends naturally to the setting $\frac{n_l}{n_{l'}} \to \alpha_{l,l'} \in (0,\infty)$ as $\min\{n_1,\ldots,n_L\} \to \infty$.)

Figure S8: **Exact and experimental dynamics are nearly identical for network outputs, and are similar for individual weights (Standard parameterization).** Experiment is for an MSE loss, ReLU network with 5 hidden layers of width $n = 2048$, $\eta = 0.005/2048$ $|\mathcal{D}| = 256$, $k = 1$, $\sigma_w^2 = 2.0$, and $\sigma_b^2 = 0.1$. All three panes in the first row show dynamics for a randomly selected subset of datapoints or parameters. First two panes in the second row show dynamics of loss and accuracy for training and test points agree well between original and linearized model. Bottom right pane shows the dynamics of RMSE between the two models on test points using empirical kernel.

2. The analytic NTK $\Theta$ (defined in Equation S42) is full-rank, i.e. $0 < \lambda_{\min} := \lambda_{\min}(\Theta) \leq \lambda_{\max} := \lambda_{\max}(\Theta) < \infty$. We set $\eta_{\text{critical}} = 2(\lambda_{\min} + \lambda_{\max})^{-1}$.

3. The training set $(\mathcal{X}, \mathcal{Y})$ is contained in some compact set and $x \neq \tilde{x}$ for all $x, \tilde{x} \in \mathcal{X}$.

4. The activation function $\phi$ satisfies

$$|\phi(0)|, \quad \|\phi'\|_\infty, \quad \sup_{x \neq \tilde{x}} |\phi'(x) - \phi'(\tilde{x})|/|x - \tilde{x}| < \infty. \tag{S38}$$

Assumption 2 indeed holds when $\mathcal{X} \subseteq \{x \in \mathbb{R}^{n_0} : \|x\|_2 = 1\}$ and $\phi(x)$ grows non-polynomially for large $x$ [13]. Throughout this section, we use $C > 0$ to denote some constant whose value may depend on $L$, $|\mathcal{X}|$ and $(\sigma_w^2, \sigma_b^2)$ and may change from line to line, but is always independent of $n$.

Let $\theta_t$ denote the parameters at time step $t$. We use the following short-hand

$$f(\theta_t) = f(\mathcal{X}, \theta_t) \in \mathbb{R}^{|\mathcal{X}| \times k} \tag{S39}$$

$$g(\theta_t) = f(\mathcal{X}, \theta_t) - \mathcal{Y} \in \mathbb{R}^{|\mathcal{X}| \times k} \tag{S40}$$

$$J(\theta_t) = \nabla_\theta f(\theta_t) \in \mathbb{R}^{(|\mathcal{X}|k) \times |\theta|} \tag{S41}$$

where $|\mathcal{X}|$ is the cardinality of the training set and $k$ is the output dimension of the network. The empirical and analytic NTK of the standard parameterization is defined as

$$\begin{cases} \hat{\Theta}_t & := \hat{\Theta}_t(\mathcal{X}, \mathcal{X}) = \frac{1}{n} J(\theta_t) J(\theta_t)^T \\ \Theta & := \lim_{n \to \infty} \hat{\Theta}_0 \quad \text{in} \quad \text{probability.} \end{cases} \tag{S42}$$

Note that the convergence of the empirical NTK in probability is proved rigorously in [37]. We consider the MSE loss

$$\mathcal{L}(t) = \frac{1}{2} \|g(\theta_t)\|_2^2. \tag{S43}$$

Since $f(\theta_t)$ converges in distribution to a mean zero Guassian with covariance $\mathcal{K}$, one can show that for arbitrarily small $\delta_0 > 0$, there are constants $R_0 > 0$ and $n_0$ (both may depend on $\delta_0$, $|\mathcal{X}|$ and $\mathcal{K}$) such that for every $n \geq n_0$, with probability at least $(1 - \delta_0)$ over random initialization,

$$\|g(\theta_0)\|_2 < R_0. \tag{S44}$$

The gradient descent update with learning rate $\eta$ is

$$\theta_{t+1} = \theta_t - \eta J(\theta_t)^T g(\theta_t) \tag{S45}$$

and the gradient flow equation is

$$\dot{\theta}_t = -J(\theta_t)^T g(\theta_t). \tag{S46}$$

We prove convergence of neural network training and the stability of NTK for both discrete gradient descent and gradient flow. Both proofs rely on the local lipschitzness of the Jacobian $J(\theta)$.

**Lemma 1** (**Local Lipschitzness of the Jacobian**). *There is a $K > 0$ such that for every $C > 0$, with high probability over random initialization (w.h.p.o.r.i.) the following holds*

$$\begin{cases} \frac{1}{\sqrt{n}}\|J(\theta) - J(\tilde{\theta})\|_F & \leq K\|\theta - \tilde{\theta}\|_2 \\[2mm] \frac{1}{\sqrt{n}}\|J(\theta)\|_F & \leq K \end{cases} \quad , \quad \forall \theta, \tilde{\theta} \in B(\theta_0, Cn^{-\frac{1}{2}}) \tag{S47}$$

*where*

$$B(\theta_0, R) := \{\theta : \|\theta - \theta_0\|_2 < R\}. \tag{S48}$$

The following are the main results of this section.

**Theorem G.1** (**Gradient descent**). *Assume **Assumptions [1-4]**. For $\delta_0 > 0$ and $\eta_0 < \eta_{\text{critical}}$, there exist $R_0 > 0$, $N \in \mathbb{N}$ and $K > 1$, such that for every $n \geq N$, the following holds with probability at least $(1 - \delta_0)$ over random initialization when applying gradient descent with learning rate $\eta = \frac{\eta_0}{n}$,*

$$\begin{cases} \|g(\theta_t)\|_2 \leq \left(1 - \frac{\eta_0 \lambda_{\min}}{3}\right)^t R_0 \\[3mm] \sum_{j=1}^t \|\theta_j - \theta_{j-1}\|_2 \leq \frac{\eta_0 K R_0}{\sqrt{n}} \sum_{j=1}^t (1 - \frac{\eta_0 \lambda_{\min}}{3})^{j-1} \leq \frac{3K R_0}{\lambda_{\min}} n^{-\frac{1}{2}} \end{cases} \tag{S49}$$

*and*

$$\sup_t \|\hat{\Theta}_0 - \hat{\Theta}_t\|_F \leq \frac{6K^3 R_0}{\lambda_{\min}} n^{-\frac{1}{2}}. \tag{S50}$$

**Theorem G.2** (**Gradient Flow**). *Assume **Assumptions[1-4]**. For $\delta_0 > 0$, there exist $R_0 > 0$, $N \in \mathbb{N}$ and $K > 1$, such that for every $n \geq N$, the following holds with probability at least $(1 - \delta_0)$ over random initialization when applying gradient flow with "learning rate" $\eta = \frac{\eta_0}{n}$*

$$\begin{cases} \|g(\theta_t)\|_2 \leq e^{-\frac{\eta_0 \lambda_{\min}}{3} t} R_0 \\[3mm] \|\theta_t - \theta_0\|_2 \leq \frac{3K R_0}{\lambda_{\min}} (1 - e^{-\frac{1}{3}\eta_0 \lambda_{\min} t}) n^{-\frac{1}{2}} \end{cases} \tag{S51}$$

*and*

$$\sup_t \|\hat{\Theta}_0 - \hat{\Theta}_t\|_F \leq \frac{6K^3 R_0}{\lambda_{\min}} n^{-\frac{1}{2}}. \tag{S52}$$

See the following two subsections for the proof.

**Remark 1.** *One can extend the results in Theorem G.1 and Theorem G.2 to other architectures or functions as long as*

1. *The empirical NTK converges in probability and the limit is positive definite.*

2. *Lemma 1 holds, i.e. the Jacobian is locally Lipschitz.*

## G.1 Proof of Theorem G.1

As discussed above, there exist $R_0$ and $n_0$ such that for every $n \geq n_0$, with probability at least $(1 - \delta_0/10)$ over random initialization,

$$\|g(\theta_0)\|_2 < R_0. \tag{S53}$$

Let $C = \frac{3KR_0}{\lambda_{\min}}$ in Lemma 1. We first prove Equation S49 by induction. Choose $n_1 > n_0$ such that for every $n \geq n_1$ Equation S47 and Equation S53 hold with probability at least $(1 - \delta_0/5)$ over random initialization. The $t = 0$ case is obvious and we assume Equation S49 holds for $t = t$. Then by induction and the second estimate of Equation S47

$$\|\theta_{t+1} - \theta_t\|_2 \leq \eta \|J(\theta_t)\|_{\mathrm{op}} \|g(\theta_t)\|_2 \leq \frac{K\eta_0}{\sqrt{n}} \left( 1 - \frac{\eta_0 \lambda_{\min}}{3} \right)^t R_0, \tag{S54}$$

which gives the first estimate of Equation S49 for $t+1$ and which also implies $\|\theta_j - \theta_0\|_2 \leq \frac{3KR_0}{\lambda_{\min}} n^{-\frac{1}{2}}$ for $j = 0, \ldots, t+1$. To prove the second one, we apply the mean value theorem and the formula for gradient decent update at step $t + 1$

$$\|g(\theta_{t+1})\|_2 = \|g(\theta_{t+1}) - g(\theta_t) + g(\theta_t)\|_2 \tag{S55}$$

$$= \|J(\tilde{\theta}_t)(\theta_{t+1} - \theta_t) + g(\theta_t)\|_2 \tag{S56}$$

$$= \| - \eta J(\tilde{\theta}_t) J(\theta_t)^T g(\theta_t) + g(\theta_t)\|_2 \tag{S57}$$

$$\leq \|1 - \eta J(\tilde{\theta}_t) J(\theta_t)^T\|_{\mathrm{op}} \|g(\theta_t)\|_2 \tag{S58}$$

$$\leq \|1 - \eta J(\tilde{\theta}_t) J(\theta_t)^T\|_{\mathrm{op}} \left( 1 - \frac{\eta_0 \lambda_{\min}}{3} \right)^t R_0, \tag{S59}$$

where $\tilde{\theta}_t$ is some linear interpolation between $\theta_t$ and $\theta_{t+1}$. It remains to show with probability at least $(1 - \delta_0/2)$,

$$\|1 - \eta J(\tilde{\theta}_t) J(\theta_t)^T\|_{\mathrm{op}} \leq 1 - \frac{\eta_0 \lambda_{\min}}{3}. \tag{S60}$$

This can be verified by Lemma 1. Because $\hat{\Theta}_0 \to \Theta$ [37] in probability, one can find $n_2$ such that the event

$$\|\Theta - \hat{\Theta}_0\|_F \leq \frac{\eta_0 \lambda_{\min}}{3} \tag{S61}$$

has probability at least $(1 - \delta_0/5)$ for every $n \geq n_2$. The assumption $\eta_0 < \frac{2}{\lambda_{\min} + \lambda_{\max}}$ implies

$$\|1 - \eta_0 \Theta\|_{\mathrm{op}} \leq 1 - \eta_0 \lambda_{\min}. \tag{S62}$$

Thus

$$\|1 - \eta J(\tilde{\theta}_t) J(\theta_t)^T\|_{\mathrm{op}} \tag{S63}$$

$$\leq \|1 - \eta_0 \Theta\|_{\mathrm{op}} + \eta_0 \|\Theta - \hat{\Theta}_0\|_{\mathrm{op}} + \eta \|J(\theta_0) J(\theta_0)^T - J(\tilde{\theta}_t) J(\theta_t)^T\|_{\mathrm{op}} \tag{S64}$$

$$\leq 1 - \eta_0 \lambda_{\min} + \frac{\eta_0 \lambda_{\min}}{3} + \eta_0 K^2 (\|\theta_t - \theta_0\|_2 + \|\tilde{\theta}_t - \theta_0\|_2) \tag{S65}$$

$$\leq 1 - \eta_0 \lambda_{\min} + \frac{\eta_0 \lambda_{\min}}{3} + 2\eta_0 K^2 \frac{3KR_0}{\lambda_{\min}} \frac{1}{\sqrt{n}} \leq 1 - \frac{\eta_0 \lambda_{\min}}{3} \tag{S66}$$

with probability as least $(1 - \delta_0/2)$ if

$$n \geq \left( \frac{18K^3 R_0}{\lambda_{\min}^2} \right)^2. \tag{S67}$$

Therefore, we only need to set

$$N = \max \left\{ n_0, n_1, n_2, \left( \frac{18K^3 R_0}{\lambda_{\min}^2} \right)^2 \right\}. \tag{S68}$$

To verify Equation S50, notice that

$$\|\hat{\Theta}_0 - \hat{\Theta}_t\|_F = \frac{1}{n}\|J(\theta_0)J(\theta_0)^T - J(\theta_t)J(\theta_t)^T\|_F \tag{S69}$$

$$\leq \frac{1}{n}\left(\|J(\theta_0)\|_{\text{op}}\|J(\theta_0)^T - J(\theta_t)^T\|_F + \|J(\theta_t) - J(\theta_0)\|_{\text{op}}\|J(\theta_t)^T\|_F\right) \tag{S70}$$

$$\leq 2K^2\|\theta_0 - \theta_t\|_2 \tag{S71}$$

$$\leq \frac{6K^3 R_0}{\lambda_{\min}}\frac{1}{\sqrt{n}}, \tag{S72}$$

where we have applied the second estimate of Equation S49 and Equation S47.

## G.2  Proof of Theorem G.2

The first step is the same. There exist $R_0$ and $n_0$ such that for every $n \geq n_0$, with probability at least $(1 - \delta_0/10)$ over random initialization,

$$\|g(\theta_0)\|_2 < R_0. \tag{S73}$$

Let $C = \frac{3KR_0}{\lambda_{\min}}$ in Lemma 1. Using the same arguments as in Section G.1, one can show that there exists $n_1$ such that for all $n \geq n_1$, with probability at least $(1 - \delta_0/10)$

$$\frac{1}{n}J(\theta)J(\theta)^T \succ \frac{1}{3}\lambda_{\min}\mathbf{Id} \quad \forall \theta \in B(\theta_0, Cn^{-\frac{1}{2}}) \tag{S74}$$

Let

$$t_1 = \inf\left\{t : \|\theta_t - \theta_0\|_2 \geq \frac{3KR_0}{\lambda_{\min}}n^{-\frac{1}{2}}\right\} \tag{S75}$$

We claim $t_1 = \infty$. If not, then for all $t \leq t_1$, $\theta_t \in B(\theta_0, Cn^{-\frac{1}{2}})$ and

$$\hat{\Theta}_t \succ \frac{1}{3}\lambda_{\min}\mathbf{Id}. \tag{S76}$$

Thus

$$\frac{d}{dt}\left(\|g(t)\|_2^2\right) = -2\eta_0 g(t)^T\hat{\Theta}_t g(t) \leq -\frac{2}{3}\eta_0\lambda_{\min}\|g(t)\|_2^2 \tag{S77}$$

and

$$\|g(t)\|_2^2 \leq e^{-\frac{2}{3}\eta_0\lambda_{\min}t}\|g(0)\|_2^2 \leq e^{-\frac{2}{3}\eta_0\lambda_{\min}t}R_0^2. \tag{S78}$$

Note that

$$\frac{d}{dt}\|\theta_t - \theta_0\|_2 \leq \left\|\frac{d}{dt}\theta_t\right\|_2 = \frac{\eta_0}{n}\|J(\theta_t)g(t)\|_2 \leq \eta_0 KR_0 e^{-\frac{1}{3}\eta_0\lambda_{\min}t}n^{-1/2} \tag{S79}$$

which implies, for all $t \leq t_1$

$$\|\theta_t - \theta_0\|_2 \leq \frac{3KR_0}{\lambda_{\min}}(1 - e^{-\frac{1}{3}\eta_0\lambda_{\min}t})n^{-\frac{1}{2}} \leq \frac{3KR_0}{\lambda_{\min}}(1 - e^{-\frac{1}{3}\eta_0\lambda_{\min}t_1})n^{-\frac{1}{2}} < \frac{3KR_0}{\lambda_{\min}}n^{-\frac{1}{2}}. \tag{S80}$$

This contradicts to the definition of $t_1$ and thus $t_1 = \infty$. Note that Equation S78 is the same as the first equation of Equation S51.

## G.3  Proof of Lemma 1

The proof relies on upper bounds of operator norms of random Gaussian matrices.

**Theorem G.3** (Corollary 5.35 [48])**.** *Let $A = A_{N,n}$ be an $N \times n$ random matrix whose entries are independent standard normal random variables. Then for every $t \geq 0$, with probability at least $1 - 2\exp(-t^2/2)$ one has*

$$\sqrt{N} - \sqrt{n} - t \leq \lambda_{\min}(A) \leq \lambda_{\max}(A) \leq \sqrt{N} + \sqrt{n} + t. \tag{S81}$$

For $l \geq 1$, let

$$\delta^l(\theta, x) := \nabla_{h^l(\theta,x)} f^{L+1}(\theta, x) \in \mathbb{R}^{kn} \tag{S82}$$

$$\delta^l(\theta, \mathcal{X}) := \nabla_{h^l(\theta,\mathcal{X})} f^{L+1}(\theta, \mathcal{X}) \in \mathbb{R}^{(k \times |\mathcal{X}|) \times (n \times \mathcal{X})} \tag{S83}$$

Let $\theta = \{W^l, b^l\}$ and $\tilde{\theta} = \{\tilde{W}^l, \tilde{b}^l\}$ be any two points in $B(\theta_0, \frac{C}{\sqrt{n}})$. By the above theorem and the triangle inequality, w.h.p. over random initialization,

$$\|W^1\|_{\mathrm{op}}, \quad \|\tilde{W}^1\|_{\mathrm{op}} \leq 3\sigma_\omega \frac{\sqrt{n}}{\sqrt{n_0}}, \quad \|W^l\|_{\mathrm{op}}, \quad \|\tilde{W}^l\|_{\mathrm{op}} \leq 3\sigma_\omega \quad \text{for} \quad 2 \leq l \leq L+1 \tag{S84}$$

Using this and the assumption on $\phi$ Equation S38, it is not difficult to show that there is a constant $K_1$, depending on $\sigma_\omega^2, \sigma_b^2, |\mathcal{X}|$ and $L$ such that with high probability over random initialization[13]

$$n^{-\frac{1}{2}}\|x^l(\theta, \mathcal{X})\|_2, \quad \|\delta^l(\theta, \mathcal{X})\|_2 \leq K_1, \tag{S85}$$

$$n^{-\frac{1}{2}}\|x^l(\theta, \mathcal{X}) - x^l(\tilde{\theta}, \mathcal{X})\|_2, \quad \|\delta^l(\theta, \mathcal{X}) - \delta^l(\tilde{\theta}, \mathcal{X})\|_2 \leq K_1 \|\tilde{\theta} - \theta\|_2 \tag{S86}$$

Lemma 1 follows from these two estimates. Indeed, with high probability over random initialization

$$\|J(\theta)\|_F^2 = \sum_l \|J(W^l)\|_F^2 + \|J(b^l)\|_F^2 \tag{S87}$$

$$= \sum_l \sum_{x \in \mathcal{X}} \|x^{l-1}(\theta, x)\delta^l(\theta, x)^T\|_F^2 + \|\delta^l(\theta, x)^T\|_F^2 \tag{S88}$$

$$\leq \sum_l \sum_{x \in \mathcal{X}} (1 + \|x^{l-1}(\theta, x)\|_F^2)\|\delta^l(\theta, x)^T\|_F^2 \tag{S89}$$

$$\leq \sum_l (1 + K_1^2 n) \sum_x \|\delta^l(\theta, x)^T\|_F^2 \tag{S90}$$

$$\leq \sum_l K_1^2(1 + K_1^2 n) \tag{S91}$$

$$\leq 2(L+1)K_1^4 n, \tag{S92}$$

and similarly

$$\|J(\theta) - J(\tilde{\theta})\|_F^2 \tag{S93}$$

$$= \sum_l \sum_{x \in \mathcal{X}} \|x^{l-1}(\theta, x)\delta^l(\theta, x)^T - x^{l-1}(\tilde{\theta}, x)\delta^l(\tilde{\theta}, x)^T\|_F^2 + \|\delta^l(\theta, x)^T - \delta^l(\tilde{\theta}, x)^T\|_F^2 \tag{S94}$$

$$\leq \left( \sum_l \left( K_1^4 n + K_1^4 n \right) + K_1^2 \right) \|\theta - \tilde{\theta}\|_2 \tag{S95}$$

$$\leq 3(L+1)K_1^4 n \|\theta - \tilde{\theta}\|_2. \tag{S96}$$

### G.4  Remarks on NTK parameterization

For completeness, we also include analogues of Theorem G.1 and Lemma 1 with NTK parameterization.

**Theorem G.4** (NTK parameterization). *Assume* **Assumptions [1-4]**. *For $\delta_0 > 0$ and $\eta_0 < \eta_{\mathrm{critical}}$, there exist $R_0 > 0$, $N \in \mathbb{N}$ and $K > 1$, such that for every $n \geq N$, the following holds with probability at least $(1 - \delta_0)$ over random initialization when applying gradient descent with learning rate $\eta = \eta_0$,*

$$\begin{cases} \|g(\theta_t)\|_2 \leq \left(1 - \frac{\eta_0 \lambda_{\min}}{3}\right)^t R_0 \\ \\ \sum_{j=1}^t \|\theta_j - \theta_{j-1}\|_2 \leq K\eta_0 \sum_{j=1}^t (1 - \frac{\eta_0 \lambda_{\min}}{3})^{j-1} R_0 \leq \frac{3KR_0}{\lambda_{\min}} \end{cases} \tag{S97}$$

*and*

$$\sup_t \|\hat{\Theta}_0 - \hat{\Theta}_t\|_F \leq \frac{6K^3 R_0}{\lambda_{\min}} n^{-\frac{1}{2}}. \tag{S98}$$

**Lemma 2** (NTK parameterization: Local Lipschitzness of the Jacobian). *There is a $K > 0$ such that for every $C > 0$, with high probability over random initialization the following holds*

$$\begin{cases} \|J(\theta) - J(\tilde{\theta})\|_F & \leq K\|\theta - \tilde{\theta}\|_2 \\ \\ \|J(\theta)\|_F & \leq K \end{cases}, \qquad \forall \theta, \tilde{\theta} \in B(\theta_0, C) \tag{S99}$$

# H   Bounding the discrepancy between the original and the linearized network: MSE loss

We provide the proof for the gradient flow case. The proof for gradient descent can be obtained similarly. To simplify the notation, let $g^{\text{lin}}(t) \equiv f_t^{\text{lin}}(\mathcal{X}) - \mathcal{Y}$ and $g(t) \equiv f_t(\mathcal{X}) - \mathcal{Y}$. The theorem and proof apply to both standard and NTK parameterization. We use the notation $\lesssim$ to hide the dependence on uninteresting constants.

**Theorem H.1.** *Same as in Theorem G.2. For every $x \in \mathbb{R}^{n_0}$ with $\|x\|_2 \leq 1$, for $\delta_0 > 0$ arbitrarily small, there exist $R_0 > 0$ and $N \in \mathbb{N}$ such that for every $n \geq N$, with probability at least $(1 - \delta_0)$ over random initialization,*

$$\sup_t \left\|g^{lin}(t) - g(t)\right\|_2, \quad \sup_t \left\|g^{lin}(t,x) - g(t,x)\right\|_2 \lesssim n^{-\frac{1}{2}} R_0^2. \tag{S100}$$

*Proof.*

$$\frac{d}{dt}\left(\exp(\eta_0 \hat{\Theta}_0 t)(g^{\text{lin}}(t) - g(t))\right) \tag{S101}$$

$$= \eta_0\left(\hat{\Theta}_0 \exp(\eta_0 \hat{\Theta}_0 t)(g^{\text{lin}}(t) - g(t)) + \exp(\eta_0 \hat{\Theta}_0 t)(-\hat{\Theta}_0 g^{\text{lin}}(t) + \hat{\Theta}_t g(t))\right) \tag{S102}$$

$$= \eta_0\left(\exp(\eta_0 \hat{\Theta}_0 t)(\hat{\Theta}_t - \hat{\Theta}_0)g(t)\right) \tag{S103}$$

Integrating both sides and using the fact $g^{\text{lin}}(0) = g(0)$,

$$(g^{\text{lin}}(t) - g(t)) = -\int_0^t \eta_0\left(\exp(\eta_0\hat{\Theta}_0(s-t))(\hat{\Theta}_s - \hat{\Theta}_0)(g^{\text{lin}}(s) - g(s))\right)ds \tag{S104}$$

$$+ \int_0^t \eta_0\left(\exp(\eta_0\hat{\Theta}_0(s-t))(\hat{\Theta}_s - \hat{\Theta}_0)g^{\text{lin}}(s)\right)ds \tag{S105}$$

Let $\lambda_0 > 0$ be the smallest eigenvalue of $\hat{\Theta}_0$ (with high probability $\lambda_0 > \frac{1}{3}\lambda_{\min}$). Taking the norm gives

$$\|g^{\text{lin}}(t) - g(t)\|_2 \leq \eta_0\Big(\int_0^t \|\exp(\hat{\Theta}_0\eta_0(s-t))\|_{op}\|(\hat{\Theta}_s - \hat{\Theta}_0)\|_{op}\|g^{\text{lin}}(s) - g(s)\|_2 ds \tag{S106}$$

$$+ \int_0^t \|\exp(\hat{\Theta}_0\eta_0(s-t))\|_{op}\|(\hat{\Theta}_s - \hat{\Theta}_0)\|_{op}\|g^{\text{lin}}(s)\|_2 ds\Big) \tag{S107}$$

$$\leq \eta_0\Big(\int_0^t e^{\eta_0\lambda_0(s-t)}\|(\hat{\Theta}_s - \hat{\Theta}_0)\|_{op}\|g^{\text{lin}}(s) - g(s)\|_2 ds \tag{S108}$$

$$+ \int_0^t e^{\eta_0\lambda_0(s-t)}\|(\hat{\Theta}_s - \hat{\Theta}_0)\|_{op}\|g^{\text{lin}}(s)\|_2 ds\Big) \tag{S109}$$

Let

$$u(t) \equiv e^{\lambda_0\eta_0 t}\|g^{\text{lin}}(t) - g(t)\|_2 \tag{S110}$$

$$\alpha(t) \equiv \eta_0\int_0^t e^{\lambda_0\eta_0 s}\|(\hat{\Theta}_s - \hat{\Theta}_0)\|_{op}\|g^{\text{lin}}(s)\|_2 ds \tag{S111}$$

$$\beta(t) \equiv \eta_0\|(\hat{\Theta}_t - \hat{\Theta}_0)\|_{op} \tag{S112}$$

The above can be written as

$$u(t) \leq \alpha(t) + \int_0^t \beta(s)u(s)ds \tag{S113}$$

Note that $\alpha(t)$ is non-decreasing. Applying an integral form of the Grönwall's inequality (see Theorem 1 in [38]) gives

$$u(t) \leq \alpha(t) \exp\left(\int_0^t \beta(s)ds\right) \tag{S114}$$

Note that

$$\|g^{\text{lin}}(t)\|_2 = \|\exp\left(-\eta_0\hat{\Theta}_0 t\right) g^{\text{lin}}(0)\|_2 \leq \|\exp\left(-\eta_0\hat{\Theta}_0 t\right)\|_{op}\|g^{\text{lin}}(0)\|_2 = e^{-\lambda_0\eta_0 t}\|g^{\text{lin}}(0)\|_2 . \tag{S115}$$

Then

$$\|g^{\text{lin}}(t) - g(t)\|_2 \leq \eta_0 e^{-\lambda_0\eta_0 t} \int_0^t e^{\lambda_0\eta_0 s}\|\hat{\Theta}_s - \hat{\Theta}_0\|_{op}\|g^{\text{lin}}(s)\|_2 ds \exp\left(\int_0^t \eta_0\|\hat{\Theta}_s - \hat{\Theta}_0\|_{op}ds\right) \tag{S116}$$

$$\leq \eta_0 e^{-\lambda_0\eta_0 t}\|g^{\text{lin}}(0)\|_2 \int_0^t \|(\hat{\Theta}_s - \hat{\Theta}_0)\|_{op}ds \exp\left(\int_0^t \eta_0\|\hat{\Theta}_s - \hat{\Theta}_0\|_{op}ds\right) \tag{S117}$$

Let $\sigma_t = \sup_{0 \leq s \leq t} \|\hat{\Theta}_s - \hat{\Theta}_0\|_{op}$. Then

$$\|g^{\text{lin}}(t) - g(t)\|_2 \lesssim \left(\eta_0 t\sigma_t e^{-\lambda_0\eta_0 t + \sigma_t\eta_0 t}\right)\|g^{\text{lin}}(0)\|_2 \tag{S118}$$

As it is proved in Theorem G.1, for every $\delta_0 > 0$, with probability at least $(1 - \delta_0)$ over random initialization,

$$\sup_t \sigma_t \leq \sup_t \|\hat{\Theta}_0 - \hat{\Theta}_t\|_F \lesssim n^{-1/2}R_0 \to 0 \tag{S119}$$

when $n_1 = \cdots = n_L = n \to \infty$. Thus for large $n$ and any polynomial $P(t)$ (we use $P(t) = t$ here)

$$\sup_t e^{-\lambda_0\eta_0 t + \sigma_t\eta_0 t}\eta_0 P(t) = \mathcal{O}(1) \tag{S120}$$

Therefore

$$\sup_t \|g^{\text{lin}}(t) - g(t)\|_2 \lesssim \sup_t \sigma_t R_0 \lesssim n^{-1/2}R_0^2 \to 0 , \tag{S121}$$

as $n \to \infty$.

Now we control the discrepancy on a test point $x$. Let $y$ be its true label. Similarly,

$$\frac{d}{dt}\left(g^{\text{lin}}(t, x) - g(t, x)\right) = -\eta_0\left(\hat{\Theta}_0(x, \mathcal{X}) - \hat{\Theta}_t(x, \mathcal{X})\right)g^{\text{lin}}(t) + \eta_0\hat{\Theta}_t(x, \mathcal{X})(g(t) - g^{\text{lin}}(t)). \tag{S122}$$

Integrating over $[0, t]$ and taking the norm imply

$$\left\|g^{\text{lin}}(t, x) - g(t, x)\right\|_2 \tag{S123}$$

$$\leq \eta_0 \int_0^t \left\|\hat{\Theta}_0(x, \mathcal{X}) - \hat{\Theta}_s(x, \mathcal{X})\right\|_2 \|g^{\text{lin}}(s)\|_2 ds + \eta_0 \int_0^t \|\hat{\Theta}_s(x, \mathcal{X})\|_2\|g(s) - g^{\text{lin}}(s)\|_2 ds \tag{S124}$$

$$\leq \eta_0\|g^{\text{lin}}(0)\|_2 \int_0^t \left\|\hat{\Theta}_0(x, \mathcal{X}) - \hat{\Theta}_s(x, \mathcal{X})\right\|_2 e^{-\eta_0\lambda_0 s}ds \tag{S125}$$

$$+ \eta_0 \int_0^t (\|\hat{\Theta}_0(x, \mathcal{X})\|_2 + \|\hat{\Theta}_s(x, \mathcal{X}) - \hat{\Theta}_0(x, \mathcal{X})\|_2)\|g(s) - g^{\text{lin}}(s)\|_2 ds \tag{S126}$$

Figure S9: **Kernel convergence.** Kernels computed from randomly initialized $\mathrm{ReLU}$ networks with one and three hidden layers converge to the corresponding analytic kernel as width $n$ and number of Monte Carlo samples $M$ increases. Colors indicate averages over different numbers of Monte Carlo samples.

Similarly, Lemma 1 implies

$$\sup_t \left\| \hat{\Theta}_0(x, \mathcal{X}) - \hat{\Theta}_t(x, \mathcal{X}) \right\|_2 \lesssim n^{-\frac{1}{2}} R_0 \tag{S127}$$

This gives

$$(S125) \lesssim n^{-\frac{1}{2}} R_0^2. \tag{S128}$$

Using Equation S118 and Equation S119,

$$(S126) \lesssim \|\hat{\Theta}_0(x, \mathcal{X})\|_2 \int_0^t \left( \eta_0 s \sigma_s e^{-\lambda_0 \eta_0 s + \sigma_s \eta_0 s} \right) \|g^{\mathrm{lin}}(0)\|_2 dt \lesssim n^{-\frac{1}{2}} . \tag{S129}$$

$\square$

# I Convergence of empirical kernel

As in Novak et al. [7], we can use Monte Carlo estimates of the tangent kernel (Equation 4) to probe convergence to the infinite width kernel (analytically computed using Equations S26, S29). For simplicity, we consider random inputs drawn from $\mathcal{N}(0, 1)$ with $n_0 = 1024$. In Figure S9, we observe convergence as both width $n$ increases and the number of Monte Carlo samples $M$ increases. For both NNGP and tangent kernels we observe $\|\hat{\Theta}^{(n)} - \Theta\|_F = \mathcal{O}\left(1/\sqrt{n}\right)$ and $\|\hat{\mathcal{K}}^{(n)} - \mathcal{K}\|_F = \mathcal{O}\left(1/\sqrt{n}\right)$, as predicted by a CLT in Daniely et al. [16].

# J Details on Wide Residual Network

Table S1: **Wide Residual Network architecture from Zagoruyko and Komodakis [14].** In the residual block, we follow Batch Normalization-ReLU-Conv ordering.

| group name | output size | block type | |
|---|---|---|---|
| conv1 | $32 \times 32$ | $[3{\times}3, \text{channel size}]$ | |
| conv2 | $32 \times 32$ | $\begin{bmatrix} 3 \times 3, & \text{channel size} \\ 3 \times 3, & \text{channel size} \end{bmatrix}$ | $\times\, N$ |
| conv3 | $16 \times 16$ | $\begin{bmatrix} 3 \times 3, & \text{channel size} \\ 3 \times 3, & \text{channel size} \end{bmatrix}$ | $\times\, N$ |
| conv4 | $8 \times 8$ | $\begin{bmatrix} 3 \times 3, & \text{channel size} \\ 3 \times 3, & \text{channel size} \end{bmatrix}$ | $\times\, N$ |
| avg-pool | $1 \times 1$ | $[8 \times 8]$ | |

Figure S10: **Kernel convergence.** Kernels from single hidden layer randomly initialized ReLU network convergence to analytic kernel using Monte Carlo sampling ($M$ samples). See §I for additional discussion.

## Footnotes

[12]Combining the usual two stage update into a single equation.

[13]These two estimates can be obtained via induction. To prove bounds relating to $x^l$ and $\delta^l$, one starts with $l = 1$ and $l = L$, respectively.