[Reviews · NeurIPS 2019]

Reviewer 1



I enjoyed reading this paper. This paper builds on the recent NTK paper and develops rather surprising theory that the gradient descent dynamics of a deep neural network is actually very close to the dynamics of a simple linearized model, for wide neural networks. I’m also very impressed by the empirical results that the paper provides, because the experimental results corroborate the theory even for realistic-sized networks. Given that NTK-related results are getting much attention these days, I believe this paper would be worth reading for many people. I vote for acceptance. Minor comments: - I guess \hat \Theta^{(n)} is not defined anywhere. Does it mean the empirical tangent kernel evaluated of a width-n network? - Line 127: There is no e^{-\eta \Theta_0 t} term in Eq 2 and 3? - Line 263: Effects of depth? Not width? The paragraph doesn’t have any discussion on depth. - Figure 4: dashed lines are almost invisible; can you try to improve readability? [After rebuttal] I have read the authors’ response and the other reviews and I think my concerns were well-addressed. I’ll keep my score.

Reviewer 2



The paper shows that the dynamics of gradient descent for optimizing an infinite width neural network can be explained by the first-order Taylor expansion of the network around its initial parameters. Furthermore, it shows that when the loss function is squared loss, the dynamics admits a closed-form solution. It also provides a learning rate threshold such that whenever the learner rate is smaller than that threshold, the trajectory of gradient descent is in a neighborhood of the trajectory of gradient descent on the linearized neural net, under the condition that the neural net is sufficiently wide. It also shows that the prediction of a neural network can be described by Gaussian Process when the width goes towards infinity. The paper is written well and the insight is very interesting. I enjoy reading the paper and I also think the contributions might be significant. Q1: The theorem requires that \Theta is full rank. Does it hold in practice? Is it a strong assumption? It looks like the authors did not check this assumption in the experiments. Q2: (line S84) Can you explain how to use Theorem G.3 to get (S84)? Typo: (S77) \Theta_t <- \hat{\Theta_t} (S96) Should there be a distance factor \| \theta - \tilde{\theta} \| ?

Reviewer 3



The paper was proofread, well-structured, and very clear. The experiments were clearly described in detail, and provided relevant results. Below we outline some detailed comments of the results. 1) Relation to "On Lazy Training in Differentiable Programming" by Chizat and Bach - Some of the main results of this paper are very similar to those proved by Chizat and Bach. In particular, Chizat and Bach prove that the training of an NTK parameterized network is closely modeled by "lazy training" (their terminology for a linearized model). This paper is not referenced in the related work section. This seriously detracts from the novelty of the submission. 2) Applicability of the proven results to modern networks - The authors claim that the NTK parameterization closely models the modern neural networks used in practice. While it is true that the 1/sqrt(n) scaling is used in many modern networks, the optimization algorithms, and in particular, the learning rates used during training may invalidate the assumptions of this paper. In the aforementioned paper by Chizat and Bach they present and cite empirical evidence that the linearized networks do not model modern networks. For instance: a) Modern neural networks for image tasks have layers that learn interesting (non-random) filters. b) While the Are all layers equal'' paper does show that most layers can be reinitialized, this is not true for all layers. In particular, the first layer of each residual block in a residual network cannot be reinitialized, and do show more parameter movement than the other layers. In particular, the parameters in the first layer of the network tend to move substantially in comparison to the other layers. 3)Experimental results - While experimental results were presented for the CIFAR-10 dataset and do show that NTK models track linear models well, this doesn't provide convincing evidence that NTK parameterized networks provide a good model for modern networks. In particular, the test accuracies of the wide residual networks used are far from state of the art.

[Author Response · NeurIPS 2019]

We thank the reviewers for their time and constructive feedback on the submission, which we will incorporate to improve our manuscript.

**R1**

- We will define $\hat{\Theta}^{(n)}$ for clarity. Indeed it denotes an empirical tangent kernel for width $n$ network.

- Line 118: equation reference should point to Eq. 8-11 instead of Eq. 2-3.

- Figure 4: The primary reason why the dashed lines are hard to see is that the linearized network's training dynamics match those of the nonlinear model so closely. Having said this, we agree that we should improve the clarity of the plots and will include modified versions upon revision.

**R2**

- Rank of tangent kernel. We observe that $\Theta$ is full rank. See line 515 in the Supplementary Material and Proposition 2 in (Jacot et al. 2018): under mild assumptions (all inputs have the same norm with non-polynomial activation functions), the kernel is positive definite. To confirm that the kernel is full-rank in practice, we can generate kernels and look at their spectrum. The following plots show the spectrum of the NTK of two five-layer fully-connected models on CIFAR-10 and MNIST. We find that they are positive-definite as expected.

- Derivation of (S84): Thank you for bringing the lack of clarity here to our attention. One source of confusion may stem from typos in (S81) which should read $\lambda_{min}(A) \geq \sqrt{N} - \sqrt{n} - t, \lambda_{max}(A) \leq \sqrt{N} + \sqrt{n} + t$. With this, we now describe how (S84) is obtained. For simplicity, let us assume $\sigma_w = 1$ and that $2 \leq l \leq L$ (as arguments for $l = 1$ and $l = L + 1$ are similar). For $2 \leq l \leq L$, when $\theta = \theta_0$ (i.e. at random initialization), $W_l$ are $n \times n$ random Gaussian matrices, so with high probability (Thm G3), $\|W_l\|_{op} \leq (2 + 0.5)$. For any $\theta \in B(\theta_0, C/\sqrt{n})$, by the triangle inequality, the operator norm of $W_l$ is bounded above by $(2 + 0.5 + C/\sqrt{n}) < 3$ with high probability. We have applied the fact that the operator norm of $\Delta W_l$ is bounded by its Frobenius norm, which is at most $C/\sqrt{n}$.

- We appreciate your identification of typos, which we will fix upon revision.

**R4**

- Relation to "Lazy Training": Thank you for bringing this oversight to our attention. *"A Note on Lazy Training in Supervised Differentiable Programming"* by Chizat and Bach is an important contribution and we will absolutely include a discussion of it in relation to our own work upon revision. While there is overlap with our own submission we would like to emphasize that the work of Chizat and Bach was concurrent with our own paper (similar preprint release dates). Additionally, the initial versions (arXiv V1, V2) of that work only performed experiments on one-hidden-layer networks and some of their results (e.g. Sec 2.2 in V1, V2) are restricted to single-hidden-layer networks.

- Applicability to modern networks: As the reviewer points out, some layers of modern networks may be operating far from the linearized regime. However, in many situations increasing the size of networks can lead to improved performance (e.g. EfficientNet, XLNet). If this trend continues to be monotonic in width, the infinite-width limit might indeed be relevant for well-performing architectures. In Figure 1 of (Novak & Xiao et al. 2019; `https://arxiv.org/abs/1810.05148`), it is shown that the comparison of performance between finite- and infinite-width networks is highly architecture-dependent. In particular, it was found that infinite-width networks perform as well as or better than their finite-width counterparts for many fully-connected or locally-connected architectures. Similarly, in Table 1 of (Arora et al, 2019; `https://arxiv.org/pdf/1904.11955.pdf`) NTK kernels outperform the corresponding finite width CNNs in 5 out of 10 configurations (though the best overall performance is achieved by a finite width CNN). It is still an open research question to determine what are the main factors that determine these performance gaps. In any case, we believe that examining the behavior of infinitely wide networks provides a strong basis from which to build up a systematic understanding of finite-width networks (and/or networks trained with large learning rates).

- Implications of "gradient descent doesn't generate samples from a probabilistic model": Very briefly: Infinitely-wide neural networks open up ways to study deep neural networks both under fully Bayesian training through the Gaussian Process correspondence, and under Gradient Descent (GD) training through the NTK or the linearization perspective. The resulting distributions over functions are inconsistent (the distribution resulting from GD training does not generally correspond to a Bayesian posterior). We believe understanding the biases over learned functions induced by different training schemes and architectures is a fascinating avenue for future work. We will expand discussion around this.

[Meta-Review · NeurIPS 2019]

This paper studies deep neural networks in the regime where the layer widths grow to infinity. Its main contribution is to show that the dynamics of gradient descent for optimizing an infinite width neural network can be explained by the first-order Taylor expansion of the network around its initial parameters, given by the NTK of Jacot et al. Reviewers all agreed this is a valuable contribution which helps the current efforts on understanding the inner workings of gradient descent on large neural networks and its role with regards to generalisation. Despite some concerns about the applicability of this regime to explain the empirical performance of large deep nets and some concurrent work (Chizat and Bach), the authors successfully addressed these concerns in the rebuttal and therefore the AC recommends acceptance.